# Ultrastable halide perovskite CsPbBr₃ photoanodes achieved with electrocatalytic glassy-carbon and boron-doped diamond sheets

Zhonghui Zhu[1,2], Matyas Daboczi [1], Minzhi Chen [1], Yimin Xuan [2] ✉, Xianglei Liu[2] & Salvador Eslava [1] ✉

Halide perovskites exhibit exceptional optoelectronic properties for photo-electrochemical production of solar fuels and chemicals but their instability in aqueous electrolytes hampers their application. Here we present ultrastable perovskite CsPbBr₃-based photoanodes achieved with both multifunctional glassy carbon and boron-doped diamond sheets coated with Ni nanopyramids and NiFeOOH. These perovskite photoanodes achieve record operational stability in aqueous electrolytes, preserving 95% of their initial photocurrent density for 168 h of continuous operation with the glassy carbon sheets and 97% for 210 h with the boron-doped diamond sheets, due to the excellent mechanical and chemical stability of glassy carbon, boron-doped diamond, and nickel metal. Moreover, these photoanodes reach a low water-oxidation onset potential close to +0.4 $V_{RHE}$ and photocurrent densities close to 8 mA cm⁻² at 1.23 $V_{RHE}$, owing to the high conductivity of glassy carbon and boron-doped diamond and the catalytic activity of NiFeOOH. The applied catalytic, protective sheets employ only earth-abundant elements and straightforward fabrication methods, engineering a solution for the success of halide perovskites in stable photoelectrochemical cells.

Wide utilization of fossil fuels exacerbates $CO_2$ emissions, leading to serious global warming and climate change[1–3]. Solar energy, the most abundant renewable energy source, can be directly converted into stable chemical energy in fuels such as hydrogen and hydrocarbons via photoelectrochemical (PEC) water and carbon dioxide conversion[4–6], which provides a promising way to produce clean fuels and chemicals[7–10]. The solar-to-fuel conversion efficiency is required to be over 10% for the commercialization of PEC devices[11,12]. Best performances in recent years have been achieved by engineering and tuning the PEC device light absorption, charge transport, and surface reactions[12–15]. In this respect, the oxygen evolution reaction (OER) on the photoanode side is considered the bottleneck to achieve high efficiencies, due to the sluggish kinetics to form $O_2$ involving 4 holes per molecule. Developing efficient, stable, and low-cost photoanodes remains a challenge to meeting the criteria for commercializing PEC devices.

The last 10 years have seen the emergence of halide perovskite materials as one of the most promising semiconductor candidates for photovoltaics because of their broad light absorption, long charge lifetime, and low cost, translated into considerable power conversion efficiency currently reaching values above 26%[16–20]. However, halide perovskite materials suffer instability problems in aqueous conditions that limit their direct application in PEC cells. This is especially critical for photoanodes that run under harsher oxidizing conditions[21,22].

[1]Department of Chemical Engineering and Centre for Processable Electronics, Imperial College London, London SW7 2AZ, UK. [2]School of Energy and Power Engineering, Nanjing University of Aeronautics and Astronautics, Nanjing 210016, China. ✉e-mail: ymxuan@nuaa.edu.cn; s.eslava@imperial.ac.uk

To overcome the instability of halide perovskite materials, Poli et al. proposed in 2019 the use of a stack of mesoporous carbon layer, graphite sheet, and surface electrocatalyst to protect CsPbBr$_3$-based photoanodes and achieve stable solar OER for at least 34 h (without electrocatalyst)[23]. Other groups have also explored the protection of perovskite-based photoanodes with gold, Ni foil, carbon film, carbon powder, and Ni-coated graphite sheet, achieving photoanode stabilities up to tens of hours[24-30]. For example, Kim et al. fabricated a perovskite-based photoanode using a carbon conductive layer, Ni foil, and a 3D-structured Ni as protective catalyst layers[26], which showed solar OER stability of 48 h, but the photocurrent dropped by 50% of the initial performance. Recently, Daboczi et al. reported a composite protective stack consisting of low-temperature carbon layer, a dense graphite sheet, and a less-dense graphite sheet functionalized with NiFeOOH that reaches 70 h stable solar-driven OER keeping 83% of its initial photocurrent or 95% of its stabilized photocurrent[31]. The porous graphite sheet provides protection and a high surface area for electrocatalysts and reactions. However, the layered structure of porous graphite sheet suffers from chemical and mechanical stress due to the OER and nucleation and growth of bubbles, which results in the necessity to replace the graphite sheets upon signs of deterioration for operations longer than 80 h. Although various impermeable materials have been utilized to protect the perovskite layer for enhancing the stability and performance of photoanodes, few devices have been reported to achieve stable solar-driven OER operation without significant performance degradation for a couple of days. Moreover, there remains a lack of a general strategy for fabricating stable and efficient perovskite-based photoanodes.

Glassy carbon (GC) is widely used as an electrode material in electrochemistry due to its high electrical conductivity, impermeability, chemical resistance, and low cost[32,33]. GC has a much mechanically stronger fullerene-related structure than graphite, so we hypothesized that it may offer better mechanical and chemical stability under oxygen nucleation and bubble growth in oxidizing conditions[34,35]. Another carbon allotrope currently studied as electrode material with superior mechanical and chemical properties to those of graphite and GC is boron-doped diamond (BDD)[36]. Structures of the three materials (graphite, GC, and BCC) are shown in Fig. 1. Both GC and BDD materials remain unexplored for the protection of halide perovskite photoanodes, despite being impermeable, conductive, commercially available and of potential window wider than graphite.

Herein, we demonstrate the synthesis of perovskite CsPbBr$_3$-based photoanodes with impermeable GC sheet coated with Ni nanopyramids and NiFeOOH catalyst, which achieves ultrastable and efficient solar-driven OER. The as-prepared perovskite photoanodes with functionalized GC show unprecedented stability: a representative example retains 95% of its initial photocurrent density (5.8 mA cm$^{-2}$ at 1.23 V$_{RHE}$) for 168 h continuous solar-driven OER without showing signs of any significant degradation. The onset potential of this device is as low as +0.45 V$_{RHE}$ with the addition of the Ni nanopyramids and NiFeOOH catalyst electrodeposited on the GC sheets. A modular fabrication method of the perovskite-based photoanodes with protective

sheets is followed, in which solar cell devices and electrocatalyst-functionalized protection sheets are separately prepared and eventually joined by a simple spin-coated adhesive. Moreover, this strategy can also be extended to BDD sheets as the impermeable layer. The BDD-based devices achieve even higher photocurrent density and stability: a representative example maintains 97% of the initial current density (7.4 mA cm$^{-2}$ at 1.23 V$_{RHE}$) for 210 h continuous operation. To the best of our knowledge, these halide perovskite-based photoanodes achieve record operational stabilities. Our work demonstrates a general fabrication strategy of stable perovskite photoanodes, providing an effective method for durable solar energy conversion.

## Results

### Fabrication of CsPbBr$_3$ photoanodes with a protective glassy-carbon sheet

The device fabrication process of the photoanodes consisting of a photo-absorber device and a multifunctional catalytic protective sheet is represented in Fig. 2. The protective catalytic sheets consist of adhesive layer, GC sheet, nickel (Ni) nanopyramids and NiFeOOH catalyst. To prepare the protective catalytic sheets, Ni nanopyramids were first electrodeposited on the rough side of commercial GC sheets, followed by electrodeposition of NiFeOOH catalyst on the nanopyramids surface (see "Methods" for synthesis details). An adhesive suspension was then spin-coated on the back flat side of the GC sheets to create a thin adhesive layer to attach the protective catalytic sheets to the perovskite photo-absorber devices. The thickness of the adhesive layer was minimized to ensure electrical contact. The photo-absorber devices were perovskite n-i-p solar cells as shown in Fig. 2b, including fluorine-doped tin oxide (FTO)-coated glass substrate, tin oxide (SnO$_2$) electron transport layer (ETL), CsPbBr$_3$ absorber layer, and mesoporous printed carbon electrode layer. The SnO$_2$ ETL was deposited by using a simple chemical bath method[37]. The CsPbBr$_3$ layer was fabricated by a combination of PbBr$_2$ spin coating and two-step CsBr chemical bath[31]. Ultraviolet-visible (UV-vis) spectroscopy shows high light absorption of the perovskite layer (Supplementary Fig. 1b). After these steps, mesoporous carbon was blade-coated onto the perovskite layer as the conductive electrode layer. Top-view scanning electron microscopy (SEM) micrographs in Fig. 2b shows that the CsPbBr$_3$ layer offers uniform large grains (1–3 μm) and the SnO$_2$ a compact structure on the FTO substrate. X-ray diffraction (XRD) shows the presence of pure 3D CsPbBr$_3$ perovskite phase deposited on FTO/SnO$_2$[38,39] (Supplementary Fig. 1a).

To investigate the band alignment in the perovskite photoanode, the energy band diagrams of all constituent layers were measured by Kelvin probe (Fermi level/work function), ambient photoemission spectroscopy (valence band edge), and incident photon-to-current spectroscopy (band gap) (Fig. 2c and Supplementary Fig. 1c, d). The energy diagram shows that electrons and holes generated in the CsPbBr$_3$ layer would be effectively collected in the SnO$_2$ and mesoporous carbon layers, respectively. The work function of bare GC sheet layers is 4.40 eV, and it is reduced to 4.18 eV after the electrodeposition of Ni nanopyramids. They are both smaller than that of the

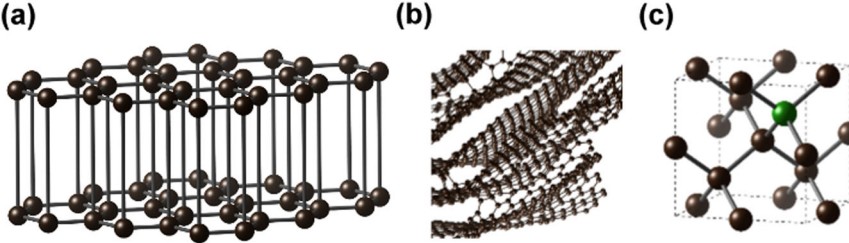

**Fig. 1 | Schematic representations of the atomic structures of different carbon allotropes. a** Graphite, (**b**) glassy carbon, and (**c**) boron-doped diamond. Black ball: carbon. Green ball: boron.

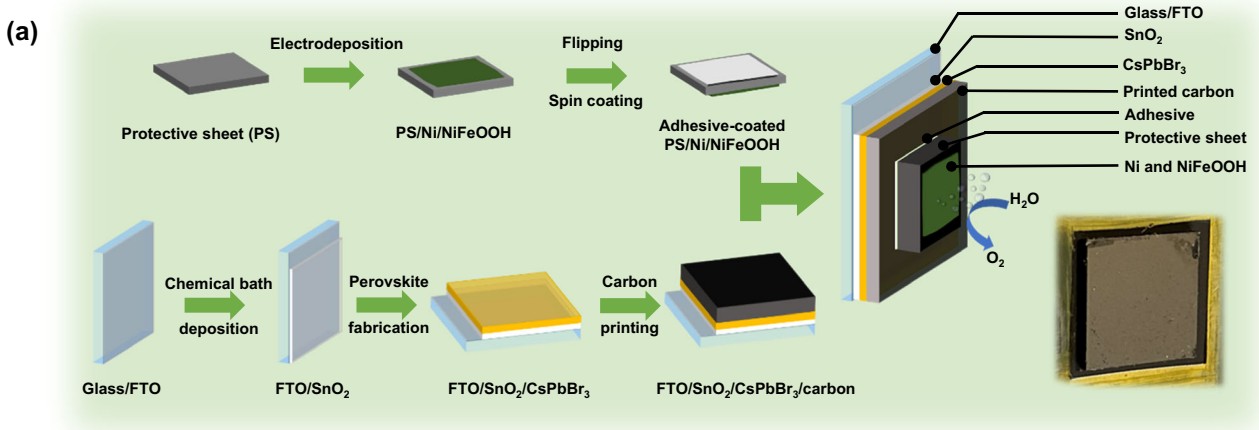

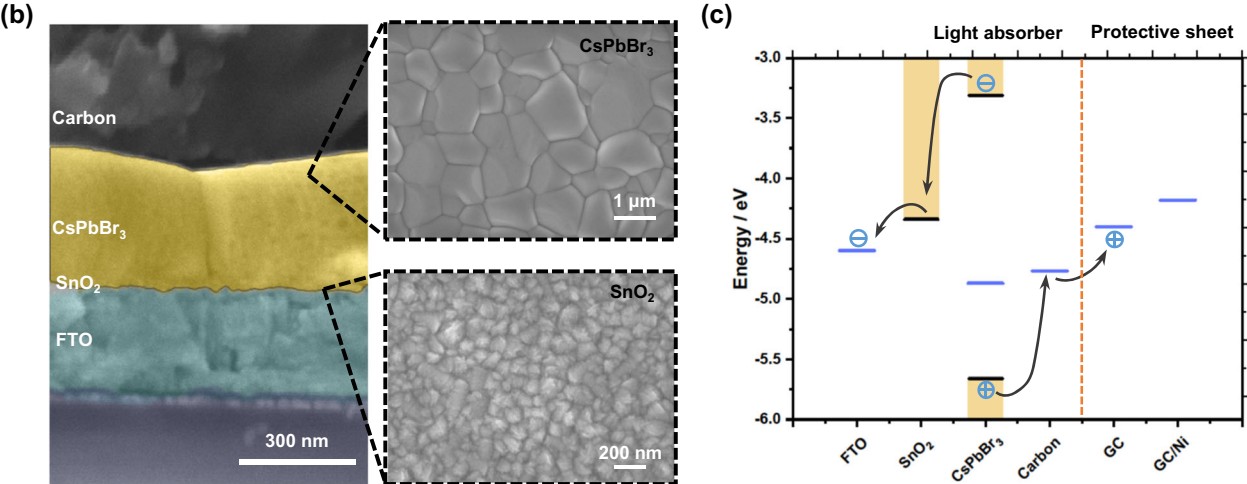

**Fig. 2 | Fabrication process of CsPbBr₃ photoanodes with protective catalytic sheet. a** Schematic diagram describing the preparation of the catalyst-coated protection sheet made of glassy carbon and the perovskite light absorber to form photoanodes. Right bottom inset: photograph of the CsPbBr₃ photoanode protected with GC/Ni/NiFeOOH. **b** Cross-sectional and top-view SEM micrographs of the as-prepared CsPbBr₃ perovskite solar cell. **c** Energy band diagrams of each component in the photoanode and illustration of charge transport.

mesoporous carbon layer (4.77 eV), providing an energetic barrier-free path for the hole transport from the perovskite to the carbon electrode and then to the surface of the protection sheet.

**Characterization and electrochemical performance of the Ni nanopyramids and NiFeOOH functionalized glassy-carbon sheet**

GC was studied as the substrate of protective sheets due to its excellent conductivity and mechanical and chemical stability. However, a bare GC sheet can be slowly oxidized by water at the high anodic potentials needed for OER[40,41], and its intrinsic OER catalytic performance is poor because of its inertness and small surface area. To counter these undesired properties, the top GC side was roughened with silicon carbide abrasive paper and then a Ni nano-structured layer was electrodeposited, both to reduce the OER onset potential and improve the reaction surface[42,43]. As shown in Fig. 3a, the Ni nanopyramids have smooth faces, clear edges, and their base covers completely the GC sheet surface. The average base and height of the pyramids are around 300–500 and 400 nm, respectively (Supplementary Fig. 2). This nanopyramid structure is expected to enhance the hole injection to the electrolyte, i.e., surface reactions. XRD patterns of the bare GC sheet and GC sheet with the Ni layer are shown in Fig. 3b. There are three diffraction peaks at (111), (200), and (220) assigned to the Ni nanopyramids, indicating a face-centered cubic phase of nickel[44].

To improve the catalytic ability of our protective sheet, an OER catalyst, NiFeOOH, was applied on the Ni nanostructure layer via a straightforward electrodeposition method[9], which can provide uniform deposition of catalyst on the Ni layer. To identify the catalytic function of Ni and NiFeOOH deposition, linear sweep voltammetry (LSV) was measured to compare the OER catalytic activities among bare GC sheet, GC/Ni, GC/NiFeOOH, and GC/Ni/NiFeOOH in a 1 M NaOH electrolyte using a three-electrode setup (Fig. 3c). The NiFeOOH and Ni electrodeposition can be successfully confirmed by energy-dispersive X-ray spectroscopy (EDS) spectra where the characteristic energies of nickel, iron and oxygen can be observed (Supplementary Figs. 3 and 4). The bare GC sheet has poor catalytic performance, thus high OER onset potential ($E_{on}$) at +2.2 $V_{RHE}$ (conservatively defined from the tangent to the maximum linear slope of current rise, Fig. 3c). After the electrodeposition of the Ni layer, GC/Ni sheets decrease the OER $E_{on}$ to +1.8 $V_{RHE}$. The small peak at +1.4 $V_{RHE}$ is assigned to Ni activation[45]. With the application of NiFeOOH, the OER $E_{on}$ of GC/Ni/NiFeOOH is further reduced to +1.55 $V_{RHE}$, which demonstrates efficient OER performance (just 0.32 V of overpotential). In addition, NiFeOOH is also electrodeposited on the bare GC sheet for a complete comparison. The OER $E_{on}$ of GC/NiFeOOH is about +1.7 $V_{RHE}$, higher than that of GC/Ni/NiFeOOH, which means that both Ni and NiFeOOH layers reduce the OER overpotential in the GC/Ni/NiFeOOH samples. These $E_{on}$ differences can be attributed to the

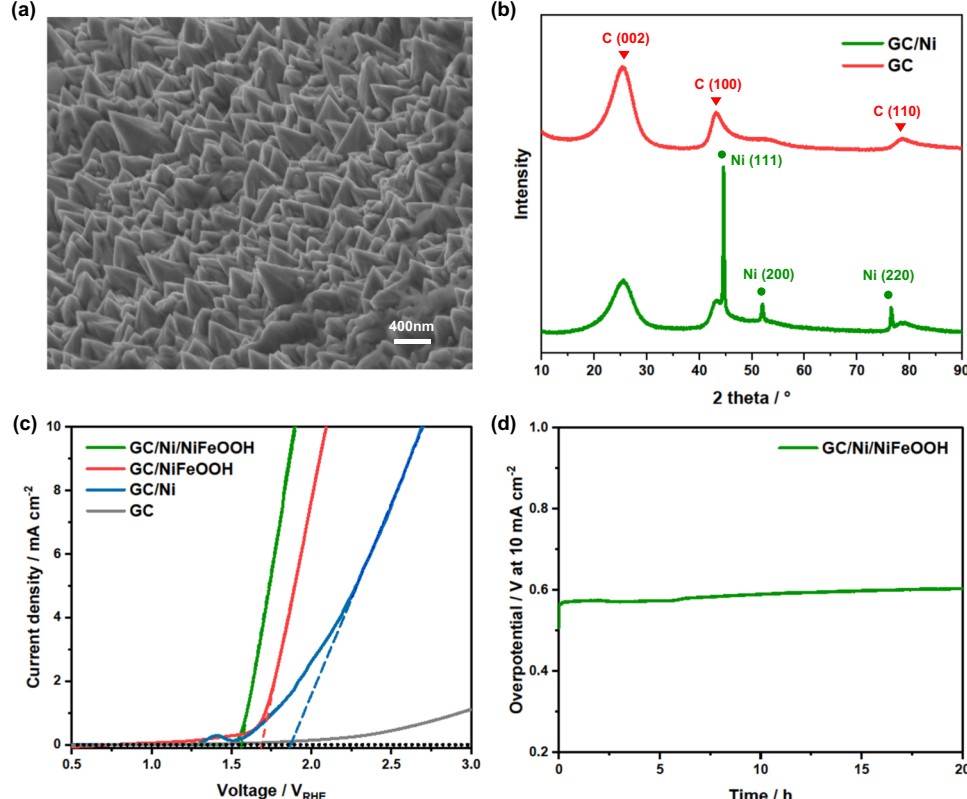

**Fig. 3 | Characterization and electrochemical performance of catalytic GC sheet. a** Tilted (45°) SEM micrograph of a GC sheet coated with Ni nanopyramids and NiFeOOH catalyst. **b** XRD diffractograms of bare GC sheet and GC/Ni sheet. **c** LSV plots (in the dark) of various functionalized GC sheets with Ni nanopyramids and NiFeOOH (solid lines) and linear extrapolation for OER onset potential determination (dashed lines). Electrolyte is aqueous 1 M NaOH (pH 14). 50 mV s$^{-1}$ scan rate. **d** Electrochemical overpotential stability test at 10 mA cm$^{-2}$ in 1 M NaOH (pH 14) on GC/Ni/NiFeOOH catalytic sheet.

larger surface reaction area provided by the Ni nanopyramids and to the catalytic activity of Ni and especially NiFeOOH. Some current on GC/NiFeOOH is observed from +1.0 to +1.5 V$_{RHE}$, which can be mainly attributed to carbon oxidation in the absence of the Ni nanopyramid coverage. Indeed, the Ni layer-coated GC sheets exhibit no current at the same potentials, indicating that the Ni layer effectively prevents GC oxidation and favors a clearcut OER $E_{on}$. The Tafel plots of the various protective sheets were also calculated, confirming the superiority of GC/Ni/NiFeOOH with a Tafel slope of 101 mV dec$^{-1}$ (Supplementary Fig. 5). The OER overpotential stability of the GC/Ni/NiFeOOH sheet for 20 h at 10 mA cm$^{-2}$ is shown in Fig. 3d and confirmed to be stable, with only a small increase over time.

## PEC performance and operational stability of CsPbBr$_3$ photoanodes with functionalized protective sheets

After successfully demonstrating the OER catalytic activity of the GC/Ni/NiFeOOH sheet, we focused on the PEC performance and stability of the CsPbBr$_3$ photoanode protected with it (Fig. 4). The protective sheet was adhered and electrically contacted to the CsPbBr$_3$ photoabsorber device by using a thin adhesive layer prepared by diluting a commercial adhesive in toluene and spin coating. To confirm the electrical contact across the interfaces, 2-electrode *J-V* scans of FTO/carbon and FTO/carbon/adhesive/GC (and later BDD) were measured, showing similar high slope values, that is, similar small resistances of only 3–4 Ω (Supplementary Fig. 6a, b). The surface of printed carbon is rough, ensuring the presence of surface pockets for the adhesive and spikes for the direct electrical connection to the GC (and later BDD) sheet, as sketched in Supplementary Fig. 6. Therefore, no conductive fillers are needed in the adhesive layer provided there is enough

roughness at the interface[29]. Moreover, the GC (and BDD) sheets are very conductive, 45 (and 10) μΩ m, so the resistance across their 1 (and 0.8) mm thickness is negligible, only 450 (and 80) μΩ per cm$^2$ sheet area. To further characterize the devices, we conducted photoluminescence (PL) measurements through the back of a sample (glass side) with all the transport layers without and with a GC (and BDD) protective sheet on top of the printed carbon layer. Due to the measurement configuration, the PL intensity is smaller than usual, and the PL spectra are less symmetric (Supplementary Fig. 6c, d). The results showed that there are no differences in PL intensity between the device without and with protective sheet on top of the printed carbon layer. Moreover, there is no PL shift. Therefore, there is negligible non-radiative recombination caused by the GC (or BDD) sheet addition, and the charge transfer between the perovskite and the protective layers is not affected. Furthermore, the photovoltaic (PV) performances of a device without and with protective sheets were compared (Supplementary Fig. 7). Practically the same performance was observed upon addition of the protective sheet, assigned to its high conductivity (resistivity of GC: 45 μΩ m, BDD: 10 μΩ m).

Current densities in the dark and under 1 sun at different applied biases are shown in Fig. 4a. A representative protected CsPbBr$_3$ photoanode exhibits a solar-driven OER $E_{on}$ of +0.5 V$_{RHE}$ and achieves a photocurrent density of 5.8 mA cm$^{-2}$ at +1.23 V$_{RHE}$. The low $E_{on}$ is the result of the low OER overpotential of GC/Ni/NiFeOOH and high photovoltage provided by the CsPbBr$_3$ photo-absorber device, in agreement with the 1.17 V of $V_{oc}$ measured as a solar cell using electrical contacts on the FTO and GC/Ni/NiFeOOH (Supplementary Fig. 8). The high photocurrent density is in good agreement with the uniform large grains structure, the deep valence band edge supplying photoinduced

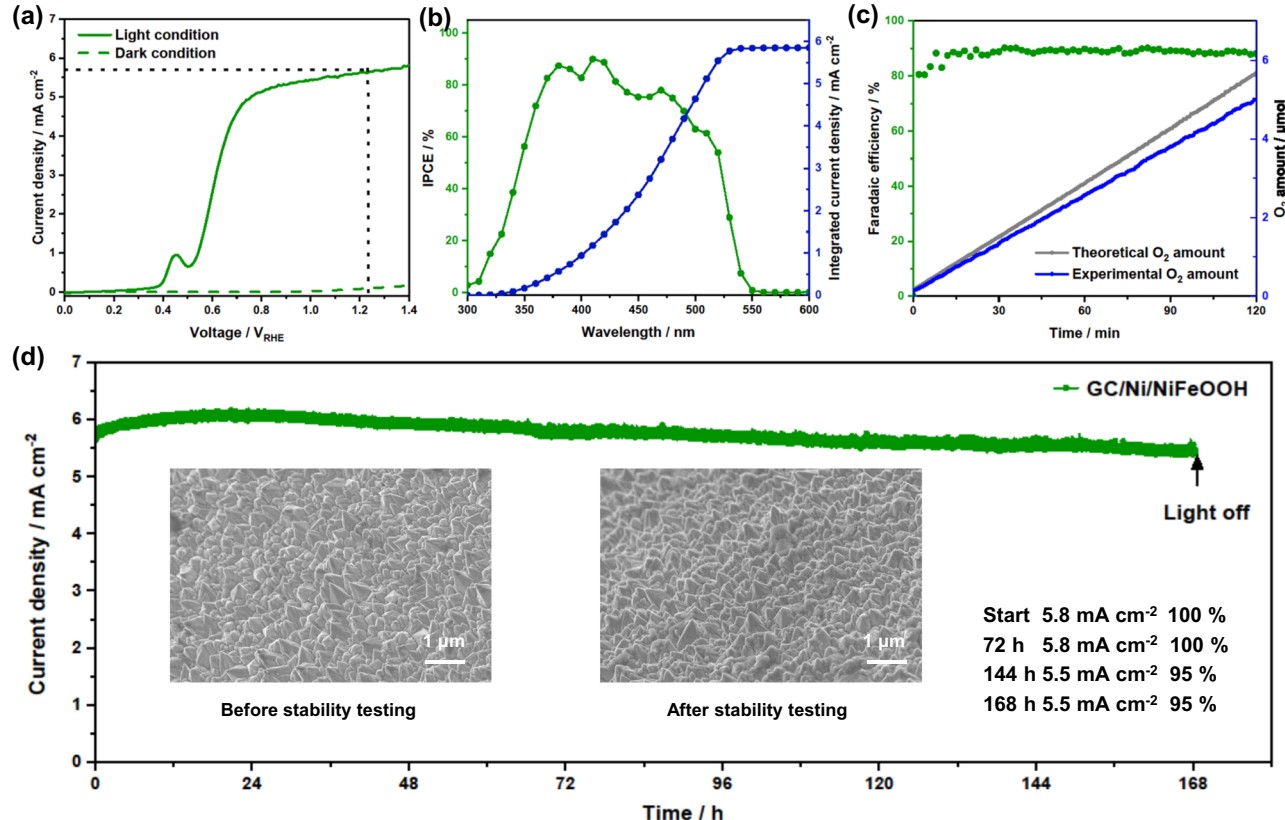

**Fig. 4 | PEC performance of CsPbBr₃ photoanodes with GC/Ni/NiFeOOH protective catalytic sheet. a** LSV polarization scan (50 mV s⁻¹ scan rate) under 1 sun illumination (solid line) and in dark (dashed green line). **b** IPCE spectrum at +1.23 V_RHE. **c** OER Faradaic efficiency calculated from the experimental O₂ amount and the theoretical O₂ amount based on the measured photocurrent. **d** Photocurrent stability measurement at +1.23 V_RHE under 1 sun illumination. Inset: SEM micrographs of Ni/NiFeOOH nanopyramid structure before and after the stability measurement and values of photocurrents at different times. Electrolyte: 1 M NaOH (pH 14).

holes with strong driving force for OER (Fig. 2c), interfacial energetics favoring charge separation (Fig. 2c), and the high light absorptance of the perovskite layer (Supplementary Fig. 1b). The applied bias photon-to-current efficiency (ABPE) was calculated to have a maximum of 2.45% at +0.7 V_RHE (Supplementary Fig. 10b). To further characterize the photoanode, the incident photon-to-current efficiency (IPCE) of the device at different wavelengths was measured at +1.23 V_RHE (Fig. 4b). The IPCE spectrum shows an onset at 550 nm, in agreement with the 2.3 eV bandgap of CsPbBr₃, and efficiencies around 80%, going down below 375 nm due to the glass support absorption. The integrated photocurrent with respect to the solar spectrum equals 5.8 mA cm⁻², in agreement with the simulated sunlight results of 5.8 mA cm⁻² (Fig. 4a). The oxygen evolution with the photocurrents is confirmed by Faradaic efficiency measurements in Fig. 4c. The average Faradaic efficiency for OER of the perovskite photoanode is about 91%, demonstrating that the photocurrent observed is ascribed to O₂ production. Deviation from 100% is assigned to diffusion or small leakage of O₂ through rubber fittings, O₂ reduction in the counter electrode, or bubble trapping at the reactor walls, resulting in a slightly lower O₂ amount detected by the sensor.

Next, the long-term stability of the GC/Ni/NiFeOOH-protected CsPbBr₃ photoanode was investigated. The photoanode was tested in a three-electrode setup at +1.23 V_RHE under 1 sun illumination. The chronoamperometry of the device for OER measurement is illustrated in Fig. 4d. The photoanode shows excellent stability, demonstrated for 168 h (8 days) with a final photocurrent density of 5.5 mA cm⁻² (95% of the initial performance). During the first 15 h, the photocurrent density slightly increases to 6.1 mA cm⁻² probably due to activation of the Ni and NiFeOOH layers under OER conditions.

From the 15 h on, the photocurrent is very stable, only decreasing 0.0037 mA cm⁻² h⁻¹. The device measured as a solar cell shows similar small decay, confirming the good translation of stability from solar cell to PEC application (Supplementary Fig. 9). The PEC workstation lamp is turned off at the 168 h for post-test characterization, but the device still works. The Ni nanopyramid structure of the protective sheet shows no change after the long 168 h stability test (inset SEM micrograph in Fig. 4d). LSV of the photoanode before and after stability test is presented in Supplementary Fig. 10a. The photocurrent density at +1.23 V_RHE shows practically the same value of 5.5 mA cm⁻² as before the stability test, but the onset curve shape shows a ca. 10 times stronger Ni oxidation peak at +0.45 V_RHE. X-ray photoelectron spectroscopy (XPS) performed before the test on the GC/Ni/NiFeOOH sheet shows strong Ni metal peak at 852.9 eV assigned to the Ni nanopyramids, but this Ni metal peak drastically decreases upon the stability test (Supplementary Fig. 10c, d)[46,47]. The increase of photocurrent for the first 15 h, the drastic decrease of XPS Ni metal peak, and the stronger Ni oxidation peak at +0.45 V_RHE after the stability test are assigned to the beneficial activation and oxidation of the surface of the Ni nanopyramids. XPS also shows that there are no obvious changes in the high-resolution XPS spectrum of the Fe 3p core level (Supplementary Fig. 10e)[48].

The GC/Ni/NiFeOOH-protected photoanode has exhibited a record stability performance to the best of our knowledge. In order to further prove the universality of this protective and catalytic booster strategy and explore its limits, we used an even more stable, inert material, boron-doped diamond (BDD), as the protective sheet instead of GC[49,50]. Devices were fabricated with BDD sheets following the same approach as that used for the GC sheets, but roughening of

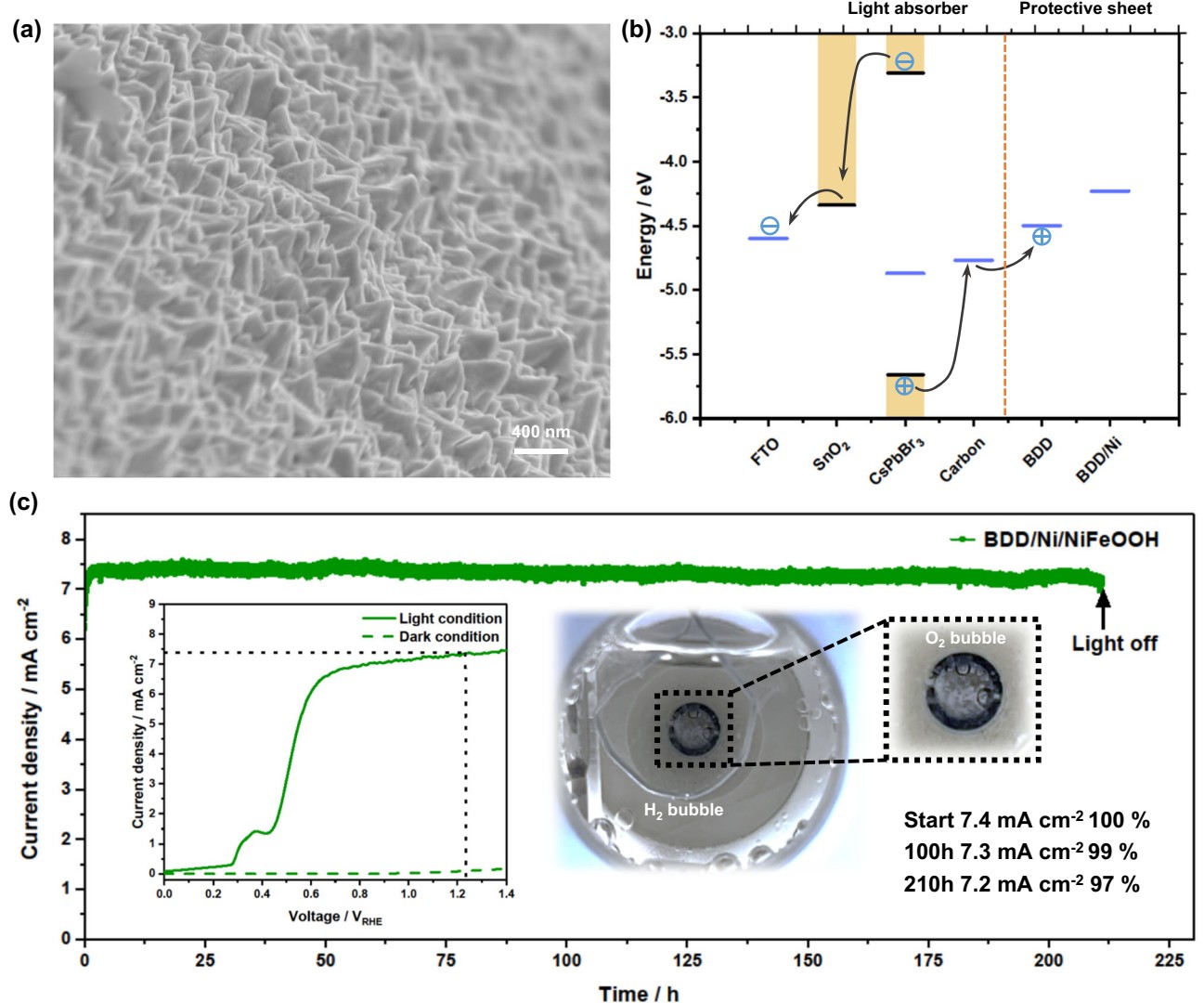

**Fig. 5 | Characterization and PEC performance of CsPbBr₃ photoanodes with protective catalytic BDD/Ni/NiFeOOH sheet. a** Tilted (45°) SEM micrograph of BDD/Ni/NiFeOOH. **b** Energy band diagram of each component in the BDD protected photoanode. **c** Main graph: stability test at +1.23 V$_{RHE}$ under 1 sun illumination. Left inset: LSV scan under 1 sun (solid line) and in dark (green dashed line) before stability test at 50 mV s⁻¹ scan rate. Right inset: photograph of the photoanode under operation showing the evolved O₂ bubbles. The electrolyte is 1 M NaOH (pH 14).

the front side was not needed because one sheet surface was already rough and because BDD is harder than the abrasive paper. The SEM micrograph of BDD/Ni/NiFeOOH sheet is shown in Fig. 5a, which shows a similar structure as that of the GC/Ni/NiFeOOH sheet. The work function of the BDD sheet and BDD/Ni sheet are measured to be −4.50 eV and −4.23 eV by Kelvin probe, respectively, which are also shallower than that of mesoporous carbon favoring charge transfer and separation (Fig. 5b and Supplementary Fig. 11). The PEC performance of the BDD/Ni/NiFeOOH-protected photoanode is illustrated in the left inset of Fig. 5c. The photoanode has a similar $E_{on}$ of +0.46 V$_{RHE}$ and achieves a higher current density (7.4 mA cm⁻² at +1.23 V$_{RHE}$), which is close to the theoretical 8.96 mA cm⁻² for the 2.3 eV bandgap of CsPbBr₃[51]. Atomic force microscopy (AFM) is carried out to compare the surface roughness of GC and BDD front sides before and after Ni/NiFeOOH electrodeposition (Supplementary Fig. 12). The average roughness $R_a$ values of the front side of GC before and after Ni and NiFeOOH deposition are 45.7 and 85.5 nm and those of BDD 125.1 and 160.4 nm, respectively. The surface area S$_a$ values of the front side of GC before and after Ni and NiFeOOH deposition are 1.017 and 1.096 m² per m² of projected area (m²m⁻²), respectively, and

those of BDD are 1.084 and 1.561 m²m⁻², respectively. Therefore, the improved performance of BDD devices compared to those of GC is in part assigned to its increased surface area, which promotes charge transfer and oxygen evolution by providing more reaction sites and interface to the electrolyte. The right inset in Fig. 5c and Supplementary Movie 1 show the O₂ bubbles produced on the reaction surface of the BDD/Ni/NiFeOOH-based photoanode. Importantly, the BDD/Ni/NiFeOOH-based photoanode also exhibits remarkable OER stability. Similar to the GC/Ni/NiFeOOH-based photoanode, there is a slight increase of photocurrent in the first 15 h due to Ni activation, followed by very stable photocurrent only decreasing 0.0010 mA cm⁻² h⁻¹. The photocurrent is stable for 210 h, achieving a final photocurrent density of 7.2 mA cm⁻² which is a remarkable 97% of the initial stabilized value (7.2 = 0.97 × 7.4). The ABPE was calculated to have a maximum of 3.84% at +0.6 V$_{RHE}$ (Supplementary Fig. 13a). Upon 210 h test, the device still works but the measurement is stopped for post-test characterization. LSV of the photoanode at the end of the test shows a very similar PEC performance to the initial one, with only changes at low potentials due to the Ni/NiFeOOH activation (Supplementary Fig. 13b).

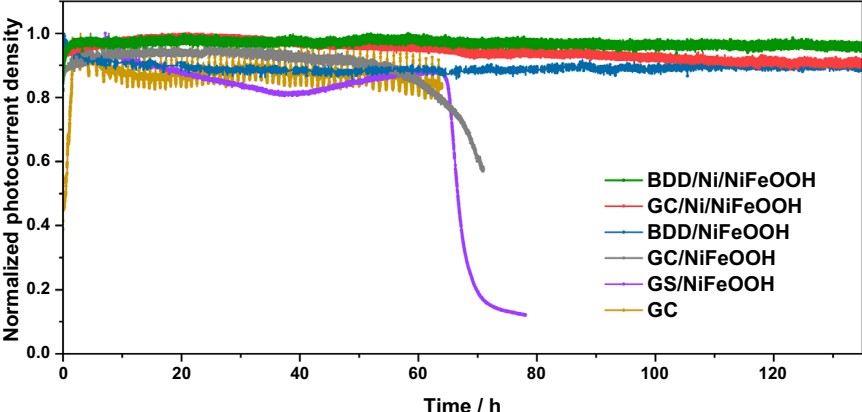

**Fig. 6 | Normalized PEC photocurrent stability test of CsPbBr$_3$ photoanodes with various protective catalytic sheets at +1.23 V$_{RHE}$ under 1 sun illumination in aqueous 1 M NaOH (pH 14).** The sample with GC was stopped because the electrolyte turned yellow in the first hours due to degradation. The one with GS/NiFeOOH was stopped at 80 h because the device had already lost 90% of its initial photocurrent. The one with GC/NiFeOOH was stopped when photocurrent reached a linear steep decay of 5% h$^{-1}$ for 1 day, which was clearly irreversible. The one with BDD/NiFeOOH lost 10% photocurrent in the first hours and remained stable beyond that. The ones with GC/Ni/NiFeOOH and BDD/Ni/NiFeOOH showed practically no degradation (just 3% and 2% upon 135 h, respectively).

## Operational stability comparison of CsPbBr$_3$ photoanodes with various protective sheets

A comparison on the stability of CsPbBr$_3$ photoanodes with different protective structures, including graphite, glassy carbon, and boron-doped diamond sheets is presented in Fig. 6. The current densities are normalized to the stabilized initial current densities in the first few minutes. Both BDD and GC sheets with Ni nanopyramids and NiFeOOH catalyst provide very stable performance with little attenuation. The photoanode with a bare GC sheet also provides a stable photocurrent, but the electrolyte becomes yellow due to GC corrosion and nano carbon particle dispersion (Supplementary Fig. 14). Moreover, its photocurrent fluctuates due to cyclic formation and desorption of relatively large O$_2$ bubbles in the absence of the spiky Ni/NiFeOOH nanopyramids. A photoanode with GC/NiFeOOH maintains stable operation for about 60 h but then drops slowly. This is ascribed to NiFeOOH delamination/loss while the GC surface gets corroded. The photoanode with BDD/NiFeOOH exhibits stable photocurrent for most of the time; however, the photocurrent decreased in the first few hours, and the onset potential and actual current density are lower than that of the device with BDD/Ni/NiFeOOH sheet. SEM analysis on the surface morphology upon stability tests for both photoanodes with GC/NiFeOOH and BDD/NiFeOOH without the Ni nanopyramids show that the GC sheet was profoundly deteriorated whereas BDD did not show appreciable changes (Supplementary Fig. 15). Finally, a photoanode with 70 µm thick graphite sheet (GS) and NiFeOOH, previous state-of-the-art protective catalytic sheet[31] shows 70 h of stable photocurrent before pronounced decrease. Moreover, this GS/NiFeOOH photoanode shows prolonged periods of increasing and decreasing photocurrent, assigned to periods of increase in surface reaction area due to swelling and NiFeOOH catalyst loss, respectively. Altogether, the comparison evidences the importance of both the Ni nanopyramids and NiFeOOH and the superiority of BDD, followed by GC, and finally GS for excellent long-term stabilities and high PEC performance. Moreover, the GC/Ni/NiFeOOH and BDD/Ni/NiFeOOH sheets can be peeled off from the photo-absorber device, cleaned with hydrochloric acid and reused to fabricate multiple photoanodes even after long stability measurements (see "Methods" for cleaning details). LSV polarization scans of 17 photoanodes fabricated by reusing 3 GC and 2 BDD sheets, respectively, are presented in Supplementary Figs. 16 and 17. The GC/Ni/NiFeOOH protected CsPbBr$_3$ photoanodes achieve an average 5.6 (±0.35) mA cm$^{-2}$ at +1.23 V$_{RHE}$ and an average +0.45 (±0.07) V solar-driven $E_{on}$. Slightly superior, the BDD/Ni/

NiFeOOH protected CsPbBr$_3$ photoanodes achieve an average 7.2 (±0.8) mA cm$^{-2}$ at +1.23 V$_{RHE}$ and an average +0.47 (±0.03) V solar-driven $E_{on}$. The narrow standard deviation of photocurrents and $E_{on}$ confirms the reproducibility of the device fabrication procedure and the robustness of the reused GC and BDD sheets, as well the good economic practicability of our electrocatalytic protective sheets for large-scale production.

A comparison of reported stabilities for halide perovskite-based devices including photoanodes is displayed in Supplementary Fig. 18. Different types of protected perovskite-based photoanodes have been reported, but most of the devices do not have stable photocurrents, due to degradation of the photoabsorber and/or the protection. They often show a significant decrease in performance during the stability test, although they can achieve a high initial ABPE due to their organic-inorganic hybrid composition and small bandgap. Details of the photoanodes are listed in Supplementary Table 1. Our all-inorganic CsPbBr$_3$ photoanodes protected with GC/Ni/NiFeOOH and especially BDD/Ni/NiFeOOH achieve record stabilities, that is record preservation of the initial photocurrent. Moreover, our approach follows an economical and scalable fabrication strategy using commercial sheets, spin coating and electrochemical deposition, as well as abundant elements (carbon, nickel, iron and oxygen), to achieve high stability and performance.

To conclude, a general and effective approach for fabricating water-resistant halide perovskite photoanodes has been demonstrated employing impermeable and catalytic sheets made of earth-abundant elements and commercially available glassy carbon and boron-doped diamond sheets, coated with Ni nanopyramids and NiFeOOH. The photo-absorber device and the protective catalytic sheet components are prepared separately, then a thin spin-coated adhesive layer is straightforwardly applied to attach both components ensuring electrical contact. The design and physical properties of the protective catalytic sheet provides both excellent mechanical and aqueous permeability resistance and high catalytic performance. When these sheets are tested on CsPbBr$_3$ halide perovskite photo-absorber devices, resulting photoanodes with functionalized impermeable sheets display long-term stability for solar-driven water oxidation. A representative glassy-carbon protected photoanode retains 95% of its initial photocurrent density (5.8 mA cm$^{-2}$ at +1.23 V$_{RHE}$) for 168 h, whereas a photoanode protected with boron-doped diamond preserves record 97% of its initial photocurrent density (7.4 mA cm$^{-2}$ at +1.23 V$_{RHE}$) for 210 h. To the best of our knowledge, these halide perovskite

photoanodes achieve the most stable photocurrents for solar-driven oxygen evolution from aqueous electrolytes. Moreover, the onset potential of these devices reaches values as low as +0.4 $V_{RHE}$. These results indicate the versatility of our developed multifunctional protection method, paving the way for exploiting the outstanding optoelectronic properties of halide perovskites for solar fuels and feedstocks in photoelectrochemical cells.

## Methods

### Photo-absorber device fabrication

Fluorine-doped tin oxide (FTO) coated glass substrates were cleaned in a Hellmanex III detergent solution for 5 min by using ultrasonication, followed by rinsing with deionized (DI) water. Then, they were ultrasonically cleaned with acetone and later with 2-propanol (IPA) before being dried with compressed air. Next, the clean substrates were UV-ozone treated for 20 min. For the deposition of $SnO_2$ ETL by chemical bath, 100 mL of precursor solution containing 275 mg of tin chloride dihydrate (Sigma-Aldrich, ≥99.99% trace metals basis), 1.25 g of urea (Sigma-Aldrich), 1250 µL of hydrochloric acid (37%, aqueous), and 25 µL of thioglycolic acid (Sigma-Aldrich, 99%) was prepared. The top of the FTO glass was covered by tape to prevent $SnO_2$ deposition where electrical contact is later done. The precursor solution and the UV-ozone-treated FTO substrates were then loaded into a staining jar and kept in an oven at 90 °C for 4 h until the pH value of the precursor solution reached 1.5. The $SnO_2$ layers were then ultrasonicated with DI water and IPA for 5 min and then dried with compressed air. Finally, the samples were annealed at 180 °C for 30 min and treated with UV-ozone for 20 min just before perovskite deposition.

The perovskite layer was fabricated by a two-step solution deposition. A 1 M lead bromide ($PbBr_2$, TCI, >98.0%) solution in N, N-dimethylformamide (DMF) was heated to 60 °C and spin-coated on pre-heated (60 °C) FTO/$SnO_2$ substrates at 2000 rpm for 30 s and annealed on a hotplate for 30 min at 60 °C. The $PbBr_2$ layers were cooled down and immersed in a cesium bromide (CsBr Alfa Aesar, 99.999% metals basis) solution (17 mg ml$^{-1}$ in methanol) and kept at 50 °C in a vertical staining jar for 5 min. Afterward, the layers were rinsed with IPA and annealed for 5 min at 250 °C. For every substrate, the CsBr immersion, drying, and annealing steps were done twice. For the top conductive electrode layer, a carbon paste (Dycotec DM-CAP-4701S) was slowly blade-coated on the $CsPbBr_3$ layer with a gap of 0.2 mm and annealed at 70 °C for 30 min.

### Protective catalytic sheet fabrication

Protective sheets consist of a substrate (GC sheet or BDD sheet), catalytic layers, and adhesive layer. Before fabricating each protective sheet, the front sides of the GC sheets (Beijing Jingke Keyi Science Instrument Co., Ltd., 10 × 10 × 1 mm, resistivity: 45 µΩ m) were roughened with a silicon carbide abrasive paper (Allied High-Tech Inc., 600 Grit) and 0.3 alumina slurry first, and later with 0.05 µm alumina slurry (Allied High-Tech Inc., No. 90-187510 and 90-187505, resp.). The sheets were sonicated in DI water for 5 min between slurries. After roughening, the sheets were cleaned by ultrasonication in a Hellmanex III detergent solution, DI water, acetone, and IPA, followed by drying with compressed air. For the electrodeposition of the Ni nanopyramids, 50 mL of aqueous precursor solution containing 24.22 g of nickel (II) sulfamate tetrahydrate ($Ni(SO_3NH_2)_2 \cdot 4H_2O$, Sigma-Aldrich, 98%) and 5.84 g of sodium chloride (NaCl, Sigma-Aldrich, 99.999% trace metals basis) was prepared and heated to 60 °C. The back side of the GC sheet was covered by tape to prevent Ni deposition and then the GC sheet was immersed in the 60 °C pre-heated precursor solution. The GC sheet was connected to a potentiostat (Compactstat., Ivium Technologies) and a Pt counter-electrode in a two-electrode setup and 10 mA cm$^{-2}$ for 10 min were applied. After the electrodeposition, the sheet was rinsed with DI water and dried in nitrogen. For the addition of NiFeOOH, the GC/Ni sheet was immersed in 50 mL of nitrogen-

purged aqueous solution containing 525.7 mg of nickel (II) sulfate hexahydrate ($NiSO_4 \cdot 6H_2O$, Sigma-Aldrich, ≥ 98%) and 139 mg of iron (II) sulfate heptahydrate ($FeSO_4 \cdot 7H_2O$, Sigma-Aldrich, ≥ 99%). Using a three-electrode setup with a Pt counter electrode and Ag/AgCl reference electrode, linear sweeps were conducted from +0.6 to +1.2 V at 20 mV s$^{-1}$ until 4.2 mC cm$^{-2}$ was deposited. The tape coated on the other side of the protective sheet was removed after the NiFeOOH electrodeposition. Then, the obtained GC/Ni/NiFeOOH sheet was rinsed in DI water and dried before use. A commercial adhesive (RS PRO Spray Adhesive, RS) was diluted in toluene and spin-coated on the back GC side of the protection sheet at 2500 rpm for 15 s. The dilution with toluene ensured the formation of a homogenous thin adhesive film upon spin coating. A volume ratio of 1:3 (adhesive:toluene) was found optimal to avoid electrical resistance while maintaining adhesion. The protective sheet was immediately attached manually to the photo-absorber device with top carbon electrode surface to complete the photoanode. For the devices with BDD sheets (CVD BDD sheet, Ningbo GH Diamond tools Co., Ltd., 10 × 10 × 0.8 mm, resistivity: 10 µΩ m, one side polished), the BDD sheet was only cleaned by ultrasonication in a Hellmanex III detergent solution, DI water, acetone, and IPA for 5 min. No roughening of the front side was needed, and this anyway was not possible with relatively softer silicon carbide abrasive paper. Ni and NiFeOOH were deposited on the rough front side and adhesive on the polished back side following the same steps as those for the GC sheets.

### Characterization

Field-emission gun FEG SEM and EDS spectra were obtained using a Zeiss Sigma 300 at 5 kV. APS was conducted by scanning UV light irradiation (4.8–6.0 eV) and extrapolating the cube root photoemission to zero (KP Technology, SKP5050). To calculate the samples' work function, contact potential difference measurements were taken using a Kelvin probe. The work function of the vibrating tip was also considered in the calculation. The tip calibration involved a recently cleaned silver reference and APS. XRD of the thin film samples at room temperature were acquired using a Malvern Panalytical XRD X'Pert PRO system (CuKα, λ = 1.54 Å). UV-vis diffuse reflectance spectroscopy (DRS) was measured with a Shimadzu UV-3000 model equipped with an integrating sphere and $BaSO_4$ as white reference. An FLS1000 (Edinburgh Instruments) photoluminescence spectrometer was used to obtain the steady-state photoluminescence (PL) spectra with a xenon arc lamp (450 W, ozone free). The excitation wavelength was set at 405 nm with a bandwidth of 8 nm. X-ray photoelectron spectroscopy (XPS) was carried out on a Thermo Fisher K-Alpha+ using a monochromated Al Kα X-ray source. Binding energies were referenced relative to adventitious carbon at 284.6 eV. AFM study was conducted using a JPK NanoWizard 4 atomic force microscope.

### (Photo)electrochemical measurements

(Photo)electrochemical measurements were carried out by using a three-electrode setup in an aqueous 1 M NaOH electrolyte, with (photo) anodes acting as the working electrode (circular mask with predefined diameter of 5 mm was used to control the active surface area 0.197 cm$^2$) against a Pt counter electrode and a KCl-saturated Ag/AgCl reference electrode. The illuminated area was also controlled to be the same as the active surface area by using a front circular mask. The data was recorded by a Compactstat. potentiostat (Ivium Technologies). The simulated sunlight was provided by a Lot Quantum Design xenon lamp with AM 1.5 G filter. The light intensity was set at 100 mW cm$^{-2}$ (1 sun) and measured by a certified photodetector (International Light Technologies SEL623). The applied potentials against the reversible hydrogen electrode ($V_{RHE}$) were calculated from the potentials applied against the Ag/AgCl reference electrode ($V_{Ag/AgCl}$) by using this equation ($V_{RHE} = V_{Ag/AgCl} + 0.0592 \times pH + 0.1976$). ABPE spectra were calculated from LSV curves under 1 sun illumination condition in a three-electrode

setup by using the equation $ABPE(\%) = (1.23 - V_{RHE}) \times (j_{light} - j_{dark}) \times (P_{light})^{-1} \times 100\%$, where $V_{RHE}$ is the applied potential in the RHE scale, and $j_{light}$ and $j_{dark}$ are the photocurrent and dark current densities, respectively. $P_{light}$ represents the incident light intensity (100 mW cm$^{-2}$).

The photoelectrodes were also characterized as solar cells with the same setup at ambient conditions, but without electrolyte and connecting electrodes on the FTO and protective sheet applying 10 mV s$^{-1}$ reverse scan. To calculate the Faradaic efficiency, the generated $O_2$ amount was detected by a Pyroscience FireStingO2 fiber-optic oxygen meter (TROXROB10 oxygen probe, TDIP temperature sensor) in a sealed cell which was purged with nitrogen for 1 h before the testing. The dissolved $O_2$ in the liquid was calculated by applying Henry's law and added to the results. To obtain IPCE spectra, photocurrents of samples produced at different wavelengths (300–600 nm) of incident light were measured using a MSH-300F LOT QuantumDesign monochromator. An International Light Technologies SEL033/U photodetector was used to quantify the monochromatic light intensity. OER stability measurements were conducted under continuous simulated 1 sun at an applied bias of +1.23 $V_{RHE}$ in 1 M NaOH electrolyte (pH 14), while vigorously stirring the electrolyte. To measure the resistance $R$ across various interfaces between FTO/MC, FTO/MC/Adhesive/GC, and FTO/MC/Adhesive/BDD, contacts were done with electrodes as shown in Supplementary Fig. 6a and J-V scan curves were measured by a Compactstat. potentiostat (Ivium Technologies). Then, Ohm's law was used to calculate the resistance as R = V (J A)$^{-1}$, where $V$ is voltage, $J$ current density, and $A$ area. The area was controlled to be 1 cm$^2$ in all the samples.

### Cleaning steps for reutilization of GC and BDD sheets

The used GC/Ni/NiFeOOH and BDD/Ni/NiFeOOH sheets were manually peeled off from the carbon top layer of the photo-absorber devices, and kept in hydrochloric acid (37%, aqueous) overnight to remove the Ni layer and NiFeOOH catalysts. Then, they were ultrasonically cleaned with Hellmanex III detergent solution, DI water, acetone, and IPA (each for 5 min). Finally, they were dried with compressed air and used to fabricate next devices. The cleanliness degree can be determined by testing their onset potentials of water oxidation, the larger the cleaner.

### Data availability

The data that support the findings of this study are openly available in the following Figshare data repository at https://doi.org/10.6084/m9.figshare.25250872.

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

## Acknowledgements
Y.X. and Z.Z. acknowledge the Basic Science Center Program for Ordered Energy Conversion of the National Natural Science Foundation of China (No.51888103 and 52488201). Z.Z. acknowledge the PhD short-term visiting program of Nanjing University of Aeronautics and Astronautics (No. ZDGB2021007). M.D. and S.E. acknowledge the funding of the UK Engineering and Physical Sciences Research Council (EPSRC) provided via grant EP/S030727/1.

## Author contributions
Z.Z. and S.E. designed the project. Z.Z. fabricated the photoelectrodes and performed the experiments. M.D. contributed to the experiments and design and carried out the APS, Kelvin probe, and PV measurements. M.C. contributed to the PV measurements. Z.Z. wrote the manuscript with the support of S.E., Y.X. and X.L. All authors contributed to analyzing and discussing the results.

## Competing interests
The authors declare no competing interests.
