## [Peer Review File · Nature Communications]

REVIEWER COMMENTS

Reviewer #1 (Remarks to the Author):

This paper shows that protecting a CsPbBr₃ solar cell with either glassy carbon or boron doped diamond can enhance its stability as a photoanode for water oxidation. This is a nice result and good stabilities have been found. The authors also show that a NiFeOOH catalysts is more effective when deposited on top of Ni pyramids.

The results are of interest and deserve to be published, but I question whether there are enough new results here for a high impact journal like NComms. The authors have previously published several papers using very similar systems and this is rather an incremental modification of the photoanodes reported in e.g Scalable All-Inorganic Halide Perovskite Photoanodes with >100 h Operational Stability Containing Earth-Abundant Materials, *Advanced Materials*, <https://doi.org/10.1002/adma.202304350>

Figure 7 makes it appear as if no perovskite photoanodes before the ones published here have stabilities over 70hrs (according to table S1). This is deeply disingenuous, two of the authors themselves have published stabilities of 110hrs for similar photoanodes (<https://doi.org/10.1002/adma.202304350>). Yang et al (*Nature Communications* volume 14, Article number: 5486 (2023)) also claim > 100hrs with UV filter. Fehr et al have shown >100hrs of unassisted water splitting using a tandem *Nature Communications* | (2023) 14:3797.

The band diagram in figure 2 suggests that the GC/Ni is sitting between -4.0 eV and -4.5 eV and the GC is at about -4.5 eV. Water oxidation is usually shown at energies between -5.0 eV and -6.0 eV (depending on the source), as drawn the GC/Ni should not be able to oxidise water?

Please show both the forward and JV reverse scans for the solar cells in the SI. Can the authors also show the cells with short circuit currents close to 8 mA (e.g. in Figure S18 photocurrents of up to 8 mA are measured, but the JV curves show maximum 7 mA?)

Figure 6 looks very strange – why was the light turned off at the different times chosen? It rather looks as if the data has been chopped just before the cells start to degrade? Otherwise, why were such random light off times selected? Why not measure all the cells to 210 hrs?

Reviewer #2 (Remarks to the Author):

The authors reported perovskite-based photoanodes combined with glassy-carbon and boron-doped diamond sheets for durable PEC water oxidation. The photoanodes protected by glassy carbon sheets (boron doped diamond sheets) with low onset potential (+ 0.4 VRHE) retains 95% (97%) of initial photocurrent density after 100 h (210h) continuous operation, due to the excellent mechanical and chemical stability of glassy carbon, boron-doped diamond, and nickel metal. They proposed the potential of perovskite photoanodes as a promising method of durable solar fuel conversion. The manuscript is well written and clear. It can be interest of broad readership of Nature Communications by taking into account the below comments.

Comment 1: Device stability is a key factor limiting the cost and economic benefits of PEC systems to achieve industrial scale. What is the key factor that contributed to the enhanced stability? It appears that the perovskite also affects device stability.

Comment 2: In General, Spiro-OMeTAD is used as the hole transport layer in perovskite solar cell. Here a printing carbon paste was used instead of the HTL, what is the function of the printing carbon layer? Is it for reducing cost?

Comment 3: In Fig. 6, the photoanode with BDD/NiFeOOH seems to be stable as well. What is the difference between the devices with BDD/NiFeOOH and BDD/Ni/NiFeOOH? The end of blue curve of photoanodes with BDD/NiFeOOH is covered by the red curve of GC/Ni/NiFeOOH, it could be revised to make it clear.

Comment 4: The stability of both devices with glassy carbon and boron doped diamond sheets are excellent. Could you evaluate the advantages of both protective sheets for large-scale preparation applications?

Comment 5: The adhesive layer was used to bond the carbon layer and the protective sheet and might affect hole transport. The result showed the conductivity was not changed too much. How to ensure the electrical conductivity?

Comment 6: In Figure 5c, photograph of the photoanode under operation is not clear to see the bubbles, it should be revised. In Supplementary Fig. 8c, the "Ni metal" is outside of the figure, it should be revised.

Reviewer #3 (Remarks to the Author):

In this manuscript, Zhu et al. demonstrated the effectiveness of the glassy carbon and boron-doped diamond sheets coated with Ni nanopyramids and NiFeOOH in protecting the metal halide perovskite photoanode in photoelectrochemical water splitting cells. The fabricated photoelectrochemical cells exhibited a record operational lifetime in the field. Nevertheless, the performance parameters of the device require more comprehensive assessment. From this reviewer's perspective, the work involved significant amounts of engineering works, but scientific findings were limited, thus limiting the novelty and significance of this work. Further measurement is needed to support the conclusion. At this stage of performance evaluation and scientific discussion provide in the manuscript, this reviewer cannot recommend this manuscript to be published in nature communication. To be considered for publication, the authors can consider revise the manuscript regarding to following concerns:

1. A good protective layer should be able to protects the photo-absorber and transport layers from the corrosive electrolytes without compromising the transportation of photogenerated charges. Although the fabricated protective sheets can provide excellent protection, the author should provide more assessment in term of performance compromise.
2. Could the authors provide the performance indicators such as the maximum applied bias photon-to-current efficiency (ACS Energy Lett. 2022, 7, 1, 320–327) or solar-to-hydrogen (STH) efficiency (Nature Communications (2023) 14:3797) of their photoelectrochemical cells?
3. The performance parameters of the perovskite solar cells were not detailly presented. Please provide the performance parameters of the perovskite solar cells including Voc, Jsc, FF and PCE, especially, for the PSCs before and after applied the protective sheets.
4. The author should provide measurement of the charge transfer between perovskite and the protective layers (e.g. photoluminescence), the conductivity measurement provided is not sufficient to confirm the efficient charge transfer.
5. Please provide more detail discussion on the origin of low onset potential achieved. Such as the effect of energy band alignment to the potential.
6. From the Supplementary Fig. 7, it seems that the fill factor of the PSC device was quite limited, is this due to the effect of the GC/Ni/NiFeOOH sheets?
7. In figure 4c, why the experimental amount of O₂ > the theoretical amount. I supposed it was a typo error.
8. The photoelectrochemical cells show a 5% decrease in the photocurrent after 168 hours. I wonder would this be due to the degradation of the catalytic sheet or the degradation of the perovskite

absorber. To clarify this, could the author provide a maximum power point tracking for the device under 1 sun conditions when working as a solar cell in ambient conditions.

9. Would the thickness of the protective sheet affect the stability and the compromise of device performance? Please provide details about the thickness of the protective sheet.

10. Please provide details of atomic ratio for the EDS result in figure 4 supplementary.

11. Supplementary fig. 6 showing the resistance measurement, I wonder why the J-V plot does not meet the origin of coordinates? This is an ohmic contact and I believe the J-V characteristic should be linear and meet (0; 0).

12. The active surface area of the electrode is 0.197 cm². How about the active area (illuminated area) of the perovskite solar cells?

13. Line 279, page 12 in the manuscript, the authors assigned the improved performance of BDD device compared to the GC is due to the higher surface roughness thus more reaction sites. Does that mean due to the increased surface area? Should the author provide a specific surface area measurement to support this point? Because increased roughness does not necessarily lead to increased surface area.

14. The remarkable stability of the cells reported in this work is clearly much higher than the previous reports. However, in order to make a reasonable assessment, please provide a comparison of the photon-to-current efficiency with previous report as well?

15. One problem with the carbon based-PSC is the scalability due to the sheet resistance, could the author demonstrate the performance of the cell with a large active area?

Response to Reviewer Comments

“Ultrastable halide perovskite CsPbBr₃ photoanodes achieved with electrocatalytic glassy-carbon and boron-doped diamond sheets” by Zhu et al.

Manuscript submitted to *Nature Communications* (Research Article, No. NCOMMS-23-50474)

We thank all the reviewers for their constructive feedback and for recognizing our manuscript's value and importance in terms of using glassy-carbon and boron-doped diamond sheets to achieve CsPbBr₃ photoanodes with ultralong stability (>200 h at >95% preserved photocurrent). We have addressed all the concerns raised by the reviewers and have revised the manuscript based on them. We thank the reviewers for helping to refine and improve our manuscript with the help of their useful suggestions. We present below our detailed, point-by-point responses to the reviewers' comments.

Reviewer #1:

This paper shows that protecting a CsPbBr₃ solar cell with either glassy carbon or boron doped diamond can enhance its stability as a photoanode for water oxidation. This is a nice result and good stabilities have been found. The authors also show that a NiFeOOH catalysts is more effective when deposited on top of Ni pyramids.

Response:

We are grateful to the reviewer for these insightful comments on our manuscript and for recognizing the outstanding stability of our CsPbBr₃ photoanodes achieved with the protection of glassy-carbon and boron-doped diamond sheets. We have carefully considered the reviewer's points and revised the manuscript accordingly to address all concerns and questions.

1. The results are of interest and deserve to be published, but I question whether there are enough new results here for a high impact journal like NComms. The authors have previously published several papers using very similar systems and this is rather an incremental modification of the photoanodes reported in e.g Scalable All-Inorganic Halide Perovskite Photoanodes with >100 h Operational Stability Containing Earth-Abundant Materials, *Advanced Materials*, <https://doi.org/10.1002/adma.202304350>

Response:

Firstly, we thank the reviewer for their interest in our work.

We have only published two articles on the topic of halide perovskite photoanodes, one in *Nat. Commun.* in 2019 and another one in *Adv. Mater.* in 2023. The *Nat. Commun.* 2019 paper covered the discovery of mesoporous carbon layers and graphite sheets to protect and create a halide perovskite photoanode. The *Adv. Mater.* 2023 paper followed up the previous work and was focused on the 3D crystal phase engineering of CsPbBr₃ for low-temperature photoanodes, the use of NiFeOOH as electrocatalyst, the scalability of the approaches, and the enhancement of photocurrents, however all of this was done by using the same graphite sheet protection as in the *Nat. Commun.* 2019 paper.

Our current manuscript submitted covers different novel topics. The manuscript presents for the first time the use of glassy carbon and boron-doped diamond instead of graphite sheets for the protection of a halide perovskite, CsPbBr₃, and the drastic enhancement of stability. These protective, catalytic sheets have never been used in combination with any semiconductor before in photoelectrodes, to the best of our knowledge. Moreover, we show the addition of novel Ni nanopyramids and NiFeOOH on these glassy carbon and boron-doped diamond sheets for further improvement of stability and

enhancement of catalytic activity. The glassy carbon, the doped-boron diamond, and the Ni nanopyramids were not used in any of our previous works (or elsewhere) and so they are all novel aspects of our current manuscript. Furthermore, we also demonstrate a straightforward deposition of an adhesive layer on the backside of the glassy carbon and boron-doped diamond sheets, to attach the sheets to the photo-absorber structure.

The glassy carbon and boron-doped diamond sheets with Ni/NiFeOOH allow orders of magnitude longer water oxidation stability compared to the graphite sheets used in our two previous papers. Figure 6 of the manuscript (see also below its old version) compares their performance. Devices with a single graphite sheet never exceed 80 h stability (devices fully degrade, unless the graphite sheet is replaced), and by 60 h they already lose 20% of their initial photocurrent density. Our novel boron-doped diamond sample only loses 3% of its initial photocurrent density upon 210 h of continuous operation (*i.e.*, it does not degrade and could be operated even longer). The glassy carbon one only loses 5% of its initial photocurrent density upon 168 h of continuous operation (also without any significant degradation). Importantly, the experiments in Figure 6 were simply ended by switching off the light.

Old Fig. 6 Normalized PEC stability of CsPbBr₃ photoanodes with various protective catalytic sheets (GC, GS/NiFeOOH, GC/NiFeOOH, BDD/NiFeOOH, GC/Ni/NiFeOOH, and BDD/Ni/NiFeOOH sheets) at +1.23 V_{RHE} under 1 sun illumination in aqueous 1 M NaOH (pH 14). GS: graphite sheet; GC: glassy carbon sheet; and BDD: boron-doped diamond sheet.

The excellent chemical and mechanical properties of glassy carbon and boron-doped diamond also avoids fluctuations of photocurrent, which is observed in graphite sheets due to their very porous, soft graphite layers that keep swelling during the photoelectrochemical operation and necessitates their replacement after 70 h of operation. Please see Figure R1 below, comparing photocurrents without normalization. One should note that a stable performance is needed in photoelectrodes and photoelectrochemical cells, since fluctuations would affect the selectivity in more complex reactions, such as glycerol oxidation to different products.

Glassy carbon and boron-doped diamond sheets are non-porous and much more chemically and mechanically stable than graphite. Therefore, they withstand the harsh anodic condition in photoanodes and the damaging nucleation and growth of bubbles.

Figure R1 a) Continuous OER photocurrent stability of CsPbBr₃ photoanodes with (a) self-adhesive graphite sheet with NiFeOOH, taken from Figure 4 of *Adv. Mater.*, b) self-adhesive glassy carbon with Ni nanopyramids and NiFeOOH, and c) with self-adhesive boron-doped diamond sheets with Ni nanopyramids and NiFeOOH.

In the new version of the manuscript, we have added further explanations and characterizations to address the Reviewers' comments which further emphasize the novelty and significance of our work (details in following responses).

We have also updated Figure 7 to make it more understandable, by readjusting the Y axis range to 0-1 and adding linear extrapolations in the background for visual aid, as shown in the following Figure (left: old figure; right: new figure):

(left: old figure; right: new figure):

Fig. 7 Comparison of normalized preserved photocurrent among PEC devices including halide perovskite photoanodes for solar-driven OER (AMI.5G, 1 sun)²³⁻³¹. Further details in the Supplementary Table 1. Linear extrapolations in the background are added for visual aid.

2. Figure 7 makes it appear as if no perovskite photoanodes before the ones published here have stabilities over 70hrs (according to table S1). This is deeply disingenuous, two of the authors themselves have published stabilities of 110hrs for similar photoanodes (<https://doi.org/10.1002/adma.202304350>). Yang et al (Nature Communications volume 14, Article number: 5486 (2023)) also claim > 100hrs with UV filter. Fehr et al have shown >100hrs of unassisted water splitting using a tandem Nature Communications | (2023) 14:3797.

Response:

We thank the Reviewer for their feedback on Figure 7. First, we would like to emphasize that a point in Figure 7 at for example 70 h and 83 % preserved photocurrent does not mean that this device could not last more than 70 h. To avoid this misunderstanding, we have added linear extrapolations in the background to all the points for visual aid (see Figure 7 in previous comment's response). We believe this visual aid makes the Figure clearer and puts more emphasis on the trends and not on the time the result was recorded.

Comparing photoelectrochemical results achieved in different labs is always difficult, and our last intention is to make a disingenuous Figure. When we compared the stability of our photoanodes with other works in Figure 7, we represented comparable measurements from different articles and included the details in Table S1 for transparency and frankness. The compared articles all report “photoelectrochemical devices with halide perovskite photoanodes for solar-driven OER”, as literally explained in the Figure 7 caption. We excluded unrelated perovskite articles, such as those using a halide perovskite photocathode and a BiVO₄ photoanode, since the focus of our work and Figure is on photoanodes and their harsher oxidative conditions. We confirm that Figure 7 includes the works cited by the reviewer, i.e., our previous two works (ref. 23 & 31), the work of Yang et al. (ref. 30), and the work of Fehr et al. (ref. 29).

We have just noticed that although Figure 7 and Table S1 report on the same articles, the reference numbers were not the same since the Supplementary Information reference list was restarting the numbering from 1. To avoid any confusion, now we have used the same referencing numbers in Manuscript and Supplementary Information (i.e., ref. 23-31).

Here below we specify how we have fairly compared our work with these articles cited by the reviewer:

- **Our previous work which showed >110 h stability (ref. 31):** Our previous work required the use of one dense and one porous graphite sheet and, importantly, the replacement of the porous one at 70 h with a fresh one, as shown in Figures R2a (Figure 4 of ref. 31). At 70 h, it was showing signs of deterioration, and, if the deterioration was allowed to continue, the water would have reached the perovskite in a few hours and degraded the device. Because the top graphite sheets are porous and soft, they keep swelling during the photoelectrochemical OER operation. As shown in Figure R2b (Figure S20 of ref. 31), a photoanode protected with one and two graphite sheets lasted around 40 and 70 h without replacement, respectively. We believe it is fairer to compare the stability of our glassy carbon and boron-doped diamond devices with graphite sheet devices *without graphite sheet replacement* halfway the experiment, since this is not done in the glassy carbon or boron-doped diamond devices reported in the submitted manuscript. This is explained in Table S1 note & that said “*Without graphite sheet replacement*”.

Figure R2 Continuous OER photocurrent stability of CsPbBr₃ photoanodes with self-adhesive graphite sheets with NiFeOOH, taken from (a) Figure 4 and (b) Figure S20 of ref. 31.

- **Yang et al. Nat. Commun. 2023, 14, 5486 (ref. 30):** Our Figure 7 and Table S1 compares our work with the results published in Figure 4b of ref. 30, here reproduced in Figure R3. The photoanode kept 43% of initial current after 60-h stability measurement under AM1.5G irradiation condition. AM1.5G irradiation condition is what we used in our work with glassy carbon or boron-doped diamond sheets, and it is what we claim in Figure 7 caption (note the words “*for solar-driven OER*”). Therefore, we reported in Figure 7 and Table S1 that Yang et al. device reached 43% at 60 h time. We agree that Yang et al. achieves better stabilities using a UV filter and a “12 h dark recovery” at time 72 h, as shown in Figure R3, but these conditions are not comparable to our work. Including these results in Figure 7 comparison would be misleading to readers. Anyway, even with the 12 h recovery and UV-filter, Yang et al. just kept 57% of preserved photocurrent at 132 h, below our 95% and 97% of preserved photocurrent at 168 and 210 h with glassy carbon and boron-doped diamond, respectively.

Figure R3 Normalized chronoamperometric measurement of FAPbBr₃ photoanode at 1.23 V_{RHE} with different filters, taken from Figure 4b of ref. 30.

- Fehr et al. Nat. Commun. 2023, 14, 3797 (ref. 29):** Figure 7 and Table S1 in the manuscript compares our work with the work shown in Figure 3a and Supplementary Fig. 12 of ref. 29, here reproduced in Figure R4. In that work, Fehr et al. shows that a tandem made of halide perovskite photocathode and photoanode preserves 40% of the initial photocurrent density after around 16 h of operation. Since this is a tandem device, we included the note \$ in the Supplementary Table 1 explaining details such as that the photocathode side alone kept 90% photocurrent after 60 h operation, so readers can infer that most degradation comes from the photoanode side (as expected due to its harsher oxidative conditions). Moreover, the authors also explained that “...The large variation in stability is a result of inter sample variability and changes in processing conditions, particularly relative humidity in atmosphere and exposure time, affecting the photoanode devices.” in Supplementary Fig. 12 caption of ref. 29. However, the Reviewer mentioned that Fehr et al. in ref. 29 also showed >100 h stability. We are aware of those results. Such stability was reported in their Figure 3a and Supplementary Fig. 12 (ref. 29). The >100 h stability is for perovskite/silicon tandem photoelectrodes where the last layer in contact with water is not a halide perovskite layer but a silicon solar cell. Protecting silicon is not comparable to protecting a halide perovskite layer. Silicon is much easier to protect than halide perovskites because it is more chemically stable, more thermally stable, and flatter. For example, in *Adv. Sust. Syst.* 2023, 7, 2300022 they use subsequent 15 min sonication in acetone, ethanol, and water to clean unprotected silicon photoanodes, something unthinkable for an unprotected halide perovskite. That silicon tandem structure is very different from our devices, so we believe it should not be included in Figure 7 and Table S1.

Figure R4 (a) Schematic representation of the unassisted PEC water-splitting system using halide perovskite PECs (b) Continuous operational photocurrent of the tandem, both taken from Figure 3a and Supplementary Fig. 12 of ref. 29.

In view of this comment, we have emphasized further the comparable conditions in the caption of Figure 7 and in Table S1, as follows:

Fig. 7 Comparison of normalized preserved photocurrent among PEC devices including halide perovskite photoanodes for solar-driven OER (AM1.5G, 1 sun)²³⁻³¹. Further details in the Supplementary Table 1. Linear extrapolations in the background are added for visual aid.

Supplementary Table 1 Comparison of reported photoelectrochemical devices including halide perovskite photoanodes for solar-driven OER (AM1.5G, 1 sun)

We have also added further explanation to the note \$ on Supplementary Table 1 for Fehr et al. ref. 29 results as follows:

\$... The authors also report perovskite/silicon tandem photoanodes protected with GS/IrO_x that achieve longer stabilities, but the last device layer on the aqueous electrolyte side is silicon, not a halide perovskite.

3. The band diagram in figure 2 suggests that the GC/Ni is sitting between -4.0 eV and -4.5 eV and the GC is at about -4.5 eV. Water oxidation is usually shown at energies between -5.0 eV and -6.0 eV (depending on the source), as drawn the GC/Ni should not be able to oxidise water?

Response:

At 1 M NaOH (pH 14) the water oxidation is favored, with an oxidation potential located at -4.9 eV vs. vacuum (-4.9 = -4.5 + 0.83 - 1.23 eV). The photoinduced holes from CsPbBr₃ come from a deeper valence band edge at -5.7 eV vs. vacuum. Carbon, glassy carbon and boron-doped diamond behave as

metals conducting the holes, so photoinduced holes transferred from the deeper CsPbBr₃ valence band edge, especially when external bias is also applied will be able to oxidize water (i.e. hydroxyls). This is mainly due to the alignment of the quasi-Fermi level of holes with the metal work functions (considering only small bending of quasi-Fermi level within the transport layers). The quasi-Fermi level splitting in perovskite (and organic) solar cells is clearly not determined by the difference between the work function of the two metal electrodes (Trends Chem., 2019, 1, 1, 49-62., J. Appl. Phys., 2003, 93, 3605–3614.), which is demonstrated by devices with above 1 V V_{oc} with similar work function electrodes (ACS Appl. Mater. Interfaces 2019, 11, 49, 45796–45804). As a consequence, under illumination the Fermi level of the top electrode will shift deeper together with the quasi-Fermi level of the perovskite, allowing for the oxidation of water (at -4.9 eV). This is further assisted by the application of an external bias, which will effectively push the metal work function deeper compared to the water oxidation potential.

In view of this comment, we have added the following to page 9:

Page 9:

The low E_{on} is the result of the low OER overpotential of GC/Ni/NiFeOOH and high photovoltage provided by the CsPbBr₃ photo-absorber device, in agreement with the 1.17 V of V_{oc} measured as a solar cell using electrical contacts on the FTO and GC/Ni/NiFeOOH (Supplementary Fig. 8). The high photocurrent density is in good agreement with the uniform large grains structure, the deep valence band edge supplying photoinduced holes with strong driving force for OER (Fig. 2c), interfacial energetics favoring charge separation (Fig. 2c), and the high light absorptance of the perovskite layer (Supplementary Fig. 1b).

4. Please show both the forward and JV reverse scans for the solar cells in the SI. Can the authors also show the cells with short circuit currents close to 8 mA (e.g. in Figure S18 photocurrents of up to 8 mA are measured, but the JV curves show maximum 7 mA?)

Response:

We have prepared a new device and measured both the forward and reverse JV scans with either a GC or BDD sheet (new Supplementary Figure 7). The open-circuit voltage of this device was slightly increased due to operational improvements. Some hysteresis is observed in agreement with literature (J. Mater. Chem. A, 2018, 6, 18677-18686). The PV hysteresis is less relevant for continuous operational studies (as herein performed in photoelectrochemical experiments under one selected applied bias condition and not scanning).

We did reach close to 8 mA cm⁻² with a few champion photoanodes, but, as shown in Fig. S16, the average photocurrent values of photoanodes in the box plots are around 6-7 mA cm⁻². We believe it is better to show representative/average samples in the manuscript. We always observe good agreement between PV short-circuit photocurrents and PEC photocurrent plateaus, provided the OER reaction is favored with a catalyst such as NiFeOOH.

The new version of the supplementary information includes this new Figure:

Supplementary Fig. 7 (a) Reverse current-voltage scans of a CsPbBr₃ device with and without protective sheets measured as a solar cell. Performance parameters are tabulated in the inset. (b) Forward and reverse current-voltage scans of the same device with either a protective GC and BDD sheet. 1 sun illumination. Scan rate 10 mV s⁻¹.

5. Figure 6 looks very strange – why was the light turned off at the different times chosen? It rather looks as if the data has been chopped just before the cells start to degrade? Otherwise, why were such random light off times selected? Why not measure all the cells to 210 hrs?

Response:

We would like to provide some context. Continuous operational stability experiments are very time consuming and continuously occupy the whole photoelectrochemical workstation, limiting us doing other experiments. Figure 6 alone took almost two months. Measuring all the cells for 210 h would take almost 3 months, while measurements up to 135 h already allow for a good comparison of the different photoanodes. The light was switched off when considerable degradation was observed or when practically no degradation was observed for days, as follows:

- The measurement of the photoanode with just GC was stopped at 65 h because already at 20 h the electrolyte had turned slightly yellow, as explained in page 15 and shown in Supplementary Figure 14.
- The one with GS/NiFeOOH was stopped at 80 h because the device had already lost 90% of its initial photocurrent.
- The one with GC/NiFeOOH was stopped when photocurrent reached a linear steep decay of $5\% \text{ h}^{-1}$ for 1 day, which was clearly irreversible.
- The one with BDD/NiFeOOH was stopped when no changes had been observed for 4 days.
- The one with GC/Ni/NiFeOOH was stopped when its photocurrent density had been constant for 5 days.
- The one with BDD/Ni/NiFeOOH was stopped when its photocurrent density had been constant for 8 days.

Overall, we still agree that the most important part of previous Figure 6 is the first 135 h that allows for the stability comparison of the different photoanodes. Since the full length of stability results for BDD/Ni/NiFeOOH and GC/Ni/NiFeOOH are also shown in Figure 4 and Figure 5, we have decided to update Figure 6 to show the results only up to 135 h, the time at which we stopped the measurement of BDD/NiFeOOH. We have now also added further clarification and context to the caption for each sample, as follows:

Fig. 6 Normalized PEC photocurrent stability test of CsPbBr₃ photoanodes with various protective catalytic sheets at +1.23 V_{RHE} under 1 sun illumination in aqueous 1 M NaOH (pH 14). The sample with GC was stopped because electrolyte turned yellow in the first hours due to degradation. The one with GS/NiFeOOH was stopped at 80 h because the device had already lost 90% of initial photocurrent. The one with GC/NiFeOOH was stopped when photocurrent reached a linear steep decay of $5\% \text{ h}^{-1}$ for 1 day, which was clearly irreversible. The one with BDD/NiFeOOH lost 10% photocurrent in the first hours and remained stable beyond that. The ones with GC/Ni/NiFeOOH and BDD/Ni/NiFeOOH showed practically no degradation (just 3% and 2% upon 135h, respectively).

Reviewer #2:

The authors reported perovskite-based photoanodes combined with glassy-carbon and boron-doped diamond sheets for durable PEC water oxidation. The photoanodes protected by glassy carbon sheets (boron doped diamond sheets) with low onset potential (+ 0.4 VRHE) retains 95% (97%) of initial photocurrent density after 100 h (210h) continuous operation, due to the excellent mechanical and chemical stability of glassy carbon, boron-doped diamond, and nickel metal. They proposed the potential of perovskite photoanodes as a promising method of durable solar fuel conversion. The manuscript is well written and clear. It can be interest of broad readership of Nature Communications by taking into account the below comments.

Response:

We thank the reviewer for appreciating the low onset potential and unprecedented operational water oxidation stabilities achieved by our perovskite photoelectrodes applying novel materials. We are grateful for the suggestions that helped to improve the manuscript.

1. Device stability is a key factor limiting the cost and economic benefits of PEC systems to achieve industrial scale. What is the key factor that contributed to the enhanced stability? It appears that the perovskite also affects device stability.

Response:

In our study, the key factor that contributed to the enhanced stability was the use of glassy carbon or boron-doped diamond functionalized with Ni nanopillars and NiFeOOH. This conclusion was possible because we chose relatively stable all-inorganic CsPbBr₃ halide perovskite instead of typically unstable hybrid organic-inorganic halide perovskites. The good temperature and humidity stability of CsPbBr₃ perovskite solar cells is known (J. Am. Chem. Soc., 2016, 138, 49, 15829–15832; ACS Appl. Mater. Interfaces, 2020, 12, 32, 36092–36101; ACS Appl. Mater. Interfaces 2019, 11, 33, 29746–29752). We agree that the perovskite can also affect device stability, but this was avoided in our careful selection of materials for unambiguous results. We also chose stable transport layers such as SnO₂ and mesoporous carbon layer and avoided air-unstable transport layers such as Spiro-OMeTAD.²⁹ With this design, we could focus on the stability enhancement by the catalytic, protective sheets and unambiguously show the superiority of glassy carbon (GC/Ni/NiFeOOH) sheets and boron-doped diamond (BDD/Ni/NiFeOOH) sheets over the state-of-the-art protective sheets made of graphite and NiFeOOH. We further discuss the stability topic in our response to Reviewer 3 comment 8.

2. In General, Spiro-OMeTAD is used as the hole transport layer in perovskite solar cell. Here a printing carbon paste was used instead of the HTL, what is the function of the printing carbon layer? Is it for reducing cost?

Response:

As explained above, expensive Spiro-OMeTAD is known to degrade in air,²⁹ and we managed to achieve good photocurrents and photovoltage with inexpensive carbon paste as top electrode. The advantages of printed carbon layer on top of the perovskite are various. Yes, it reduces costs but also easily covers a rough perovskite layer, ensures good electrical contact with the protective sheets made of graphite, glassy carbon or boron-doped diamond, and is also a hydrophobic protective layer. If used alone, it will protect the perovskite for up to 2-3 h in aqueous electrolyte.

In a previous work (ref. 31), we prepared CsPbBr₃ photoanodes with Spiro-OMeTAD HTL between the perovskite photoactive layer and the printed carbon; however, we found that the devices with HTL showed similar performance or underperformed the ones without HTL. This suggests that in our devices the main origin of non-radiative recombination losses is not at the perovskite/HTL interface, probably due to the charge transfer already being relatively efficient because of large interfacial band bending at the SnO₂/CsPbBr₃ interface. Moreover, the shallow highest occupied molecular orbital (HOMO) of Spiro-OMeTAD (-5.0 eV) compared to the deep valence band edge of CsPbBr₃ (-5.7 eV) might also limit the use of Spiro-OMeTAD as HTL in CsPbBr₃-based devices. Yang et al. in ref. 30 also made similar observations on the deep valence band edge of FAPbBr₃.

3. In Fig. 6, the photoanode with BDD/NiFeOOH seems to be stable as well. What is the difference between the devices with BDD/NiFeOOH and BDD/Ni/NiFeOOH? The end of blue curve of photoanodes with BDD/NiFeOOH is covered by the red curve of GC/Ni/NiFeOOH, it could be revised to make it clear.

Response:

We agree that the photoanode with BDD/NiFeOOH was stable too. However, the onset potentials for water oxidation on BDD/NiFeOOH are worse than on BDD/Ni/NiFeOOH. The same difference is observed between GC/Ni/NiFeOOH and GC/NiFeOOH (as shown in Fig. 3c of the manuscript). Therefore, we still recommend using both Ni and NiFeOOH on GC and BDD. This was discussed in our manuscript Page 14 as “*The photoanode with BDD/NiFeOOH exhibits stable photocurrent for most of the time; however, the photocurrent decreased in the first few hours, and the onset potential and actual current density are lower than that of the device with BDD/Ni/NiFeOOH sheet...*”). The Ni nanopillar layer on BDD surface provides more surface area for the deposition of NiFeOOH catalyst, as shown in Supplementary Fig. 12.

We thank the reviewer for pointing out the blue curve being covered in the figure. We have updated Fig. 6 to make it clearer. Please see the revised figure below.

Fig. 6 Normalized PEC stability of CsPbBr₃ photoanodes with various protective catalytic sheets at +1.23 V_{RHE} under 1 sun illumination in aqueous 1 M NaOH (pH 14). The sample with GC was stopped because electrolyte turned yellow in the first hours due to degradation. The one with GS/NiFeOOH was stopped at 80 h because the device had already lost 90% of initial photocurrent. The one with GC/NiFeOOH was stopped when photocurrent reached a linear steep decay of 5% h⁻¹ for 1 day, which was clearly irreversible. The one with BDD/NiFeOOH lost 10% photocurrent in the first hours and remained stable beyond that. The ones with GC/Ni/NiFeOOH and BDD/Ni/NiFeOOH practically showed no degradation (just 3% and 2% upon 135h, respectively).

4. The stability of both devices with glassy carbon and boron doped diamond sheets are excellent. Could you evaluate the advantages of both protective sheets for large-scale preparation applications?

Response:

We believe that the main advantages are their chemical and mechanical stability, which resulted in the excellent results obtained, as discussed/demonstrated in the manuscript. The mechanical properties of glassy carbon and boron-doped diamond (rigid like glass) also impart better mechanical properties to the final device, forming a sandwich with the glass support. Moreover, unlike graphite sheets, glassy carbon and boron-doped diamond sheets can be reused multiple times since only their surface is modified. Better chemical and mechanical properties, as well as reusability, will favor their application to large-scale devices.

We believe that large-scale preparation of photoelectrochemical devices will follow a modular approach, in which devices of a few centimeters (e.g. 50-100 cm²) will be repeated in series/grids and utilize concentrated sunlight. This approach minimizes mass transport and resistivity limitations in the glass/FTO support (Hankin et al., *Adv. Energy Mater.* 2021, 11, 2003286). Our design approach with glassy carbon and boron-doped diamond sheets coated with Ni/NiFeOOH will have no limitation in such large-scale preparation. There are already commercially available GC and BDD sheets of 100 cm².

In view of this comment, we have added the following sentence to page 15:

*...The narrow standard deviation of photocurrents and E_{on} confirms the reproducibility of the device fabrication procedure and the robustness of the reused GC and BDD sheets, as well the good economic practicability of our electrocatalytic protective sheets **for large-scale production**.*

5. The adhesive layer was used to bond the carbon layer and the protective sheet and might affect hole transport. The result showed the conductivity was not changed too much. How to ensure the electrical conductivity?

Response:

Roughness is key. The printed carbon is porous and rough. When the GC or BDD sheet is placed on top of the printed carbon, the glue can be squeezed into surface pockets and direct electrical contact can occur between the spiky printed carbon flakes and the GC or BDD sheet surface. This was represented in the Supplementary Figure 6 sketched diagram (here below for easier reference). Moreover, the dissolution of the adhesive in toluene (adhesive:toluene 1:3 vol.) ensured a thin and homogenous adhesive layer upon spin coating. Without dilution, non-homogenous layers and higher resistance values were achieved. However, too much dilution resulted in insufficient adhesion. A volumetric ratio of 1:3 was found optimal.

In view of this comment, we have rewritten a paragraph in page 9 and added some words and sentences to page 19 and Supplementary Figure 6 caption, as follows:

Page 9:

After successfully demonstrating the OER catalytic activity of the GC/Ni/NiFeOOH sheet, we focus on the PEC performance and stability of the CsPbBr₃ photoanode protected with it (Fig. 4). The protective sheet was adhered and electrically contacted to the CsPbBr₃ photo-absorber device by using a thin adhesive layer prepared by diluting a commercial adhesive in toluene and spin coating. To confirm

the electrical contact across the interfaces, 2-electrode J-V scans of FTO/carbon and FTO/carbon/adhesive/GC (and later BDD) were measured, showing similar high slope values, that is, similar small resistances of only 3-4 W (Supplementary Fig. 6). The surface of printed carbon is rough, ensuring the presence of surface pockets for the adhesive and spikes for the direct electrical connection to the GC (and later BDD) sheet, as sketched in Supplementary Fig. 6. Therefore, no conductive fillers are needed in the adhesive layer provided there is enough roughness at the interface.²⁹ Moreover, the GC (and BDD) sheets are very conductive, 45 (and 10) $\mu\Omega$ m, so the resistance across their 1 (and 0.8) mm thickness is negligible, only 450 (and 80) $\mu\Omega$.

Supplementary Fig. 6 (a) Sketch of resistance measurement across FTO/mesoporous carbon (MC), FTO/MC/adhesive (AD)/GC, and FTO/MC/AD/BDD interfaces. The printed carbon layer is rough and porous, ensuring the presence of pockets for the adhesive and spikes for the electrical contact to the GC or BDD sheets.

Page 19 Experimental:

...A commercial adhesive (RS PRO Spray Adhesive, RS) was diluted in toluene (~~volume ratio of 1 adhesive : 3 toluene~~) and spin-coated on the back GC side of the protection sheet at 2500 rpm for 15 s. The dilution with toluene ensured the formation of a homogenous thin adhesive film upon spin coating. A volume ratio of 1:3 (adhesive:toluene) was found optimal to avoid electrical resistance while maintaining adhesion. The...

6. In Figure 5c, photograph of the photoanode under operation is not clear to see the bubbles, it should be revised. In Supplementary Fig. 8c, the “Ni metal” is outside of the figure, it should be revised.

Response:

Thanks for the useful comments. We have modified both figures to the following clearer versions:

Fig. 5 (c) Main graph: stability test at +1.23 V_{RHE} under 1 sun illumination. Left inset: LSV scan under 1 sun (solid line) and in dark (green dashed line) before stability test at 50 mV s⁻¹ scan rate. Right inset: photograph of the photoanode under operation showing the evolved O₂ bubbles. The electrolyte is 1 M NaOH (pH 14).

Supplementary Fig. 10 (a) LSV under 1 sun (solid lines) or in dark (dashed line). (b) Applied bias photon-to-current efficiency (ABPE) of photoanode protected with GC/Ni/NiFeOOH (the data is calculated from the values of Fig. 4a). (c-e) XPS spectra of GC/Ni/NiFeOOH protected photoanode before and after 168 h photoanode stability measurement (presented in Fig. 3d). The peak increase at 0.45 V_{RHE} was caused by the surface activation (i.e., oxidation) of the Ni nanopillars during the stability test.

Reviewer #3:

In this manuscript, Zhu et al. demonstrated the effectiveness of the glassy carbon and boron-doped diamond sheets coated with Ni nanopyrramids and NiFeOOH in protecting the metal halide perovskite photoanode in photoelectrochemical water splitting cells. The fabricated photoelectrochemical cells exhibited a record operational lifetime in the field. Nevertheless, the performance parameters of the device require more comprehensive assessment. From this reviewer's perspective, the work involved significant amounts of engineering works, but scientific findings were limited, thus limiting the novelty and significance of this work. Further measurement is needed to support the conclusion. At this stage of performance evaluation and scientific discussion provide in the manuscript, this reviewer cannot recommend this manuscript to be published in nature communication. To be considered for publication, the authors can consider revise the manuscript regarding to following concerns:

Response:

We thank the reviewer for appreciating the record operational lifetimes and significant amount of engineering work. We, however, believe the article also shows scientific findings, novelty, and significance, since this is the first work to show the use of glassy carbon and boron-doped diamond sheets to protect a water-unstable semiconductor and to achieve record operational stabilities with a halide perovskite photoanode. We thank the Reviewer for their suggestions that helped to improve the manuscript and its fitting in *Nature Communications*.

1. A good protective layer should be able to protects the photo-absorber and transport layers from the corrosive electrolytes without compromising the transportation of photogenerated charges. Although the fabricated protective sheets can provide excellent protection, the author should provide more assessment in term of performance compromise.

Response:

We had included plenty of assessment and characterization of the performance of our devices, including:

- Energy diagrams of all the components (Figure 2c, 5b, S1c-d, S11, ...)
- J-V curves measured as a solar cell (Figure S8)
- J-V curves measured as photoanodes for solar OER (Figure 2c, 4a, S10a, S13b, S16, S17...)
- J-V of catalytic sheets for OER (Figure 3c, 3d, S5, S6...)
- Microscopies of the different components (Figure 2b, 3a, 5a, S2, S3, S4, S12, S15...)
- Anodic operational stability studies (Figure 3d, 4d, 5c, 6...)

And now, according to the Reviewer's comments, we have added more new measurements to support our conclusion, including:

- J-V curves of devices without and with GC or BDD (new Figure S7)
- Photoluminescence measurements of devices without and with GC or BDD (Figures S6c-d)
- Applied bias photo-to-current efficiency of photoanodes (new Figures S10b and S13a)
- PV stability of devices (Supplementary Figure S9)

- AFM surface area of protective sheets (Figure S12)

These characterizations and tests provide a clear picture of the performance of a halide perovskite photoanode protected with glassy carbon and boron-doped diamond sheets with Ni nanopillars and NiFeOOH. The characterization is done with rigor and detail, and clear links and comparisons to literature put in context the scientific relevance and novelty of the results.

Details are provided in the following responses to more specific comments.

2. Could the authors provide the performance indicators such as the maximum applied bias photon-to-current efficiency (ACS Energy Lett. 2022, 7, 1, 320–327) or solar-to-hydrogen (STH) efficiency (Nature Communications (2023) 14:3797) of their photoelectrochemical cells?

Response:

We have calculated the applied bias photon-to-current efficiency (ABPE) of our photoanodes with GC/Ni/NiFeOOH and BDD/Ni/NiFeOOH and added the results to the new Supplementary Figures 10b and 13a. The maximum ABPEs of photoanodes with GC/Ni/NiFeOOH and BDD/Ni/NiFeOOH are 2.45% and 3.84%, respectively. Although higher values were previously achieved with hybrid organic-inorganic halide perovskites such as MAPI, their instability under light conditions of such perovskites would have created confusion if chosen for this study. CsPbBr₃ has a relatively large bandgap of 2.3 eV, which results in lower values of ABPE; however, it was chosen for this work as a photoactive material due to its excellent stability. Readers would have struggled to distinguish the GC/Ni/NiFeOOH and BDD/Ni/NiFeOOH stability on a photoabsorber that is not stable. As explained to Reviewer 2, using relatively more stable CsPbBr₃ allowed us presenting a clear comparison between our proposed GC and BDD sheets and state-of-the-art graphite/NiFeOOH sheets.

Solar-to-hydrogen (STH) efficiency should be calculated in a two-electrode system, without any assistance of external applied bias. This would require the addition of a photocathode in a tandem configuration to generate enough photovoltage and photocurrent at operation point. Photocathodes (under reducing conditions) are typically more stable than photoanodes (under oxidation conditions). The STH conversion efficiency should take into account the entire PEC cell, including photoanode, photocathode, electrolyte, membrane, cables, etc. The development of photocathodes and an entire tandem cell is out of the scope of this manuscript that is focused on the stability of halide perovskite photoanodes under harsh oxidation conditions.

The performance of photoanodes in literature is typically compared in terms of JV curves (presented in Figure 2c, 4a, S10a, S13, S16, S17, ...) and incident photon-to-current efficiencies (IPCE). The IPCE results were presented in the manuscript Figure 4b.

In view of this comment, we have added the ABPE figure (Supplementary Fig. 10b and 13a and the following text:

Page 9:

... The applied bias photon-to-current efficiency (ABPE) was calculated to have a maximum of 2.45% at +0.7 V_{RHE} (Supplementary Fig. 10b).

Page 12:

... The ABPE was calculated to have a maximum of 3.84% at +0.6 V_{RHE} (Supplementary Fig. 13a).

Page 22:

... ABPE spectra were calculated from LSV curves under 1 sun illumination condition in a three-electrode setup by using the equation $ABPE(\%) = (1.23 - V_{RHE}) \times (j_{light} - j_{dark}) \times (P_{light})^{-1} \times 100\%$, where V_{RHE} is the applied potential in the RHE scale, and j_{light} and j_{dark} are the photocurrent and dark current densities, respectively. P_{light} represents the incident light intensity (100 mW cm^{-2}).

Supplementary Fig. 10 ... (b) Applied bias photon-to-current efficiency (ABPE) of photoanode protected with GC/Ni/NiFeOOH (the data is calculated from the values of Fig. 4a)....

Supplementary Fig. 13 (a) ABPE of photoanode protected with BDD/Ni/NiFeOOH sheet, calculated from the values of Fig. 5c. ...

3. The performance parameters of the perovskite solar cells were not detailedly presented. Please provide the performance parameters of the perovskite solar cells including V_{oc} , J_{sc} , FF and PCE, especially, for the PSCs before and after applied the protective sheets.

Response:

We thank the reviewer for the suggestion. We have made a new device and added their PSC performance parameters with just carbon, with carbon/GC, and carbon/BDD to new Supplementary

Figure 7 and Supplementary Table 1. The V_{oc} of this device was higher due to some operational improvements in the fabrication and PV measurement steps. It is clear that the addition of very conductive protective sheets had negligible effect on their PSC performance, which we further confirmed by the PL measurements (next comment's response). The relatively low FF can be attributed to the perovskite layer, which we are planning to improve in further work.

In view of this comment, we have added more detailed discussion in the manuscript (Page 9) and the following Supplementary Fig. 7 and Supplementary Table 1:

Page 9:

...Furthermore, the photovoltaic (PV) performances of a device without and with protective sheets were compared (Supplementary Figure 7). Practically the same performance was observed upon addition of the protective sheet, assigned to its high conductivity (resistivity of GC: $45 \mu\Omega m$, BDD: $10 \mu\Omega m$).

Supplementary Fig. 7 (a) Reverse current-voltage scans of a CsPbBr₃ device with and without protective sheets measured as a solar cell. Performance parameters are tabulated in the inset. (b) Forward and reverse current-voltage scans of the same device with either a protective GC and BDD sheet. 1 sun illumination. Scan rate 10 mV s^{-1} .

4. The author should provide measurement of the charge transfer between perovskite and the protective layers (e.g. photoluminescence), the conductivity measurement provided is not sufficient to confirm the efficient charge transfer.

Response:

According to the Reviewer's suggestion, we conducted photoluminescence (PL) measurements through the back of the samples (glass side) with all the transport layers without and with the protective sheets on top of the printed carbon layer. Due to the measurement configuration, the PL intensity is smaller than usual, and the PL spectra are less symmetric (new Supplementary Figure 6c-d). The results showed that there are no differences in PL intensity between the device without and with protective sheet on top of the printed carbon layer. Moreover, there is no PL shift. Therefore, there is negligible non-radiative recombination caused by the GC or BDD sheet addition on top of the carbon layer. The charge transfer between the perovskite and the protective layers is not affected.

In view of this comment, we have added more discussion in page 9 and 21 and supplementary Fig. 6:

Page 9:

...To further characterize the devices, we conducted photoluminescence (PL) measurements through the back of a sample (glass side) with all the transport layers without and with a GC (and BDD) protective sheet on top of the printed carbon layer. Due to the measurement configuration, the PL intensity is smaller than usual, and the PL spectra are less symmetric (Supplementary Figure 6c-d). The results showed that there are no differences in PL intensity between the device without and with protective sheet on top of the printed carbon layer. Moreover, there is no PL shift. Therefore, there is negligible non-radiative recombination caused by the GC (or BDD) sheet addition, and the charge transfer between the perovskite and the protective layers is not affected.

Page 21 Characterization:

An FLS1000 (Edinburgh Instruments) photoluminescence spectrometer was used to obtain steady-state photoluminescence (PL) spectra with a xenon arc lamp (450 W, ozone free). The excitation wavelength was set at 405 nm with a bandwidth of 8 nm.

Supplementary Fig. 6 ... (c-d) Photoluminescence (PL) spectra of the CsPbBr₃ photoanodes without and with GC and BDD sheets. This PL was measured from the back of the samples with all the transport layers, so the PL intensity is smaller than usual and the shape of PL is less symmetric.

5. Please provide more detail discussion on the origin of low onset potential achieved. Such as the effect of energy band alignment to the potential.

Response:

The low photoanode onset potential results from the low OER onset potential of the Ni/NiFeOOH catalyst layer and the large photovoltage provided by the CsPbBr₃ photoabsorber device.

The Ni/NiFeOOH evolves oxygen with a low onset potential at 1.55 V_{RHE} due to the good catalytic activity of NiFeOOH and the Ni nanopyramid structure that provides large surface area for both NiFeOOH deposition and posterior OER reactions (Fig. 3c and Supplementary Fig. 12). Moreover, Supplementary Fig. 5 shows the Tafel plot of different protective sheets. The Tafel slopes of GC and BDD were reduced with the addition of Ni nanopyramids and NiFeOOH, indicating a more responsive current and OER to the applied voltage upon Ni/NiFeOOH loading.

The suitable energy band structure promotes the electron-hole separation and charge transfer for the OER with the photoinduced holes (Fig. 2c).

This was explained in pages 5, 7, 9 and 12 of the manuscript as follows:

- Page 5: *...The energy diagram shows that electrons and holes generated in the CsPbBr₃ layer would be effectively collected in the SnO₂ and mesoporous carbon layers, respectively. The work function of bare GC sheet layers is 4.40 eV, and it is reduced to 4.18 eV after the electrodeposition of Ni nanopyramids...*
- Page 7: *...To counter these undesired properties, the top GC side was roughened with silicon carbide abrasive paper and then a Ni nano-structure layer was electrodeposited, both to reduce the OER onset potential and improve the reaction surface.*
- Page 7: *...With the application of NiFeOOH, the OER E_{on} of GC/Ni/NiFeOOH is further reduced to +1.55 V_{RHE}, which demonstrates efficient OER performance (just 0.32 V of overpotential). In addition, NiFeOOH is also electrodeposited on the bare GC sheet for a complete comparison. The OER E_{on} of GC/NiFeOOH is about +1.7 V_{RHE}, higher than that of GC/Ni/NiFeOOH, which means that both Ni and NiFeOOH layers reduce the OER overpotential in the GC/Ni/NiFeOOH samples. These E_{on} differences can be attributed to the larger surface reaction area provided by the Ni nanopyramids and to the catalytic activity of Ni and especially NiFeOOH. Some current on GC/NiFeOOH is observed from +1.0 to +1.5 V_{RHE}, which can be mainly attributed to carbon oxidation in the absence of the Ni nanopyramid coverage.*
- Page 9: *...A representative protected CsPbBr₃ photoanode exhibits a solar-driven OER E_{on} of +0.5 V_{RHE} and achieves a photocurrent density of 5.8 mA cm⁻² at +1.23 V_{RHE}. The low E_{on} is the result of the low OER overpotential of GC/Ni/NiFeOOH and high photovoltage provided by the CsPbBr₃ photo-absorber device, in agreement with the 1.17 V of V_{oc} measured as a solar cell using electrical contacts on the FTO and GC/Ni/NiFeOOH.*
- Page 9: *... The high photocurrent density is in good agreement with the uniform large grains structure and interfacial energetics favoring charge separation (Fig. 2) and the high light absorbance of the perovskite layer (Supplementary Fig. 1b).*
- Page 12: *...Therefore, the improved performance of BDD devices compared to those of GC is in part assigned to its increased surface area, which promotes charge transfer and oxygen evolution by providing more reaction sites and interface to the electrolyte.*

We believe that all these explanations and photoelectrochemistry literature cited in the manuscript provide a clear picture on the origin of the low onset potential for solar OER with our photoanodes. And in view of this comment, we have added the following to page 9:

Page 9:

The low E_{on} is the result of the low OER overpotential of GC/Ni/NiFeOOH and high photovoltage provided by the CsPbBr₃ photo-absorber device, in agreement with the 1.17 V of V_{oc} measured as a solar cell using electrical contacts on the FTO and GC/Ni/NiFeOOH (Supplementary Fig. 8). The high photocurrent density is in good agreement with the uniform large grains structure, the deep valence band edge supplying photoinduced holes with strong driving force for OER (Fig. 2c), interfacial energetics favoring charge separation (Fig. 2e), and the high light absorptance of the perovskite layer (Supplementary Fig. 1b).

6. From the Supplementary Fig. 7, it seems that the fill factor of the PSC device was quite limited, is this due to the effect of the GC/Ni/NiFeOOH sheets?

Response:

The GC/Ni/NiFeOOH does not limit the PSC performance because GC is very conductive, so any limited fill factor is attributed to the quality of the perovskite layer. We refer to our response to comment 3, in which we explained that we prepared a new device and measured its JV without and with GC/Ni/NiFeOOH sheet (new Supplementary Figure 7a). The fill factors of the same device without and with GC/Ni/NiFeOOH sheets were 0.62 and 0.60, respectively, which is practically the same. Therefore, the GC sheet does not limit the device.

7. In figure 4c, why the experimental amount of O₂ > the theoretical amount. I supposed it was a typo error.

Response:

Thanks for finding this typo error. We have now corrected the figure and updated it in the manuscript as follows:

Fig. 4 PEC performance of CsPbBr₃ photoanodes with GC/Ni/NiFeOOH protective catalytic sheet. (a) LSV polarization scan (50 mV s⁻¹ scan rate) under 1 sun illumination (solid line) and in dark (dashed green line). (b) IPCE spectrum at +1.23 V_{RHE}. (c) OER Faradaic efficiency calculated from the experimental O₂ amount and the theoretical O₂ amount based on the measured photocurrent. (d) Photocurrent stability measurement at +1.23 V_{RHE} under 1 sun illumination. Inset: SEM micrographs of Ni/NiFeOOH nanopyramid structure before and after the stability measurement and values of photocurrents at different times. Electrolyte: 1 M NaOH (pH 14).

8. The photoelectrochemical cells show a 5% decrease in the photocurrent after 168 hours. I wonder would this be due to the degradation of the catalytic sheet or the degradation of the perovskite absorber. To clarify this, could the author provide a maximum power point tracking for the device under 1 sun conditions when working as a solar cell in ambient conditions.

Response:

Following the advice, we have measured the PV stability of a CsPbBr₃ device with GC for 235 h. A maximum power point (MPP) measurement would constantly track MPP and adjust voltage, meaning a continuous changing of applied potential throughout the stability measurement. MPP testing condition would be different from PEC stability tests done at constant applied bias of +1.23 V_{RHE}. Therefore, we have measured the PV stability at a 0.2 V to make it most comparable to the PEC stability measurements, which were measured at 0.2 V vs Ag/AgCl electrode (+1.23 V_{RHE} at pH = 14) (new Figure S9). The device kept a stable photocurrent. The initial photocurrent density j_{ph} was 5.7 mA cm⁻² and 89% and 88% of it was maintained after 168 and 210 h of operation, respectively. This j_{ph} decrease is slightly larger than what we observed in PEC measurements (11 vs. 5%), but practically the same if we consider the maximum j_{ph} (instead of the initial j_{ph}) achieved at 15 h time in Figure 4d (11 vs. 10%). The operational PV stability of CsPbBr₃ is in agreement with literature (ACS Appl. Mater. Interfaces, 2020, 12, 32, 36092–36101; ACS Appl. Mater. Interfaces 2019, 11, 33, 29746–29752).

The PV stability results indicate that some decrease of performance can be assigned to the perovskite photoabsorber. We have also observed a small overpotential increase over time in the catalytic sheets (Figure 3d). Therefore, the small decrease in PEC activity can be assigned to both the perovskite absorber and the catalytic sheet. In both cases, the loss of performance is small, which makes them difficult to distinguish.

In view of this comment, we have added the following text and new Supplementary Fig. 9.

Manuscript page 8:

...The Tafel plots of the various protective sheets were also calculated, confirming the superiority of GC/Ni/NiFeOOH with a Tafel slope of 101 mV dec^{-1} (Supplementary Fig. 5). The OER overpotential stability of the GC/Ni/NiFeOOH sheet for 20 h at 10 mA cm^{-2} is shown in Fig. 3d and confirmed to be stable, with only a small increase over time.

Manuscript page 10-11:

Next, the long-term stability of the GC/Ni/NiFeOOH-protected CsPbBr₃ photoanode was investigated. The photoanode was tested in a three-electrode setup at $+1.23 \text{ V}_{\text{RHE}}$ under 1 sun illumination. The chronoamperometry of the device for OER measurement is illustrated in Fig. 4d. The photoanode shows excellent stability, demonstrated for 168 hours (8 days) with a final photocurrent density of 5.5 mA cm^{-2} (95 % of the initial performance). During the first 15 h, the photocurrent density slightly increases to 6.1 mA cm^{-2} probably due to activation of the Ni and NiFeOOH layers under OER conditions. From the 15 h on, the photocurrent is very stable, only decreasing $0.0037 \text{ mA cm}^{-2} \text{ h}^{-1}$. The device measured as a solar cell shows similar small decay, confirming the good translation of stability from solar cell to PEC application (Supplementary Fig. 9). The PEC workstation lamp is turned off at the 168 hour for post-test characterization, but the device still works. The Ni nanopyramid structure of the protective sheet shows no change after the long 168 h stability test (inset SEM micrograph in Fig. 4d)...

Supplementary Fig. 9 PV stability at 0.2 V voltage of a CsPbBr₃ device covered in GC sheet under 1 sun illumination. Initial photocurrent density j_{sc} : 5.7 mA cm^{-2} .

9. Would the thickness of the protective sheet affect the stability and the compromise of device performance? Please provide details about the thickness of the protective sheet.

Response:

The thicknesses of used GC and BDD sheets were 1 mm and 0.8 mm, respectively, as explained in the Experimental. These thicknesses were convenient to safely work with the sheets (roughening with abrasive paper, cleaning, spin coater and Ni/NiFeOOH electrodeposition). In terms of stability, the addition of Ni/NiFeOOH layer protects the GC or BDD surface exposed to the harsh electrolyte for solar OER conditions as well as makes the sheet catalytically active. Provided the Ni/NiFeOOH is homogeneously deposited on the GC or BDD surface as presented in the manuscript, the sheet thickness will not affect stability. However, if no Ni/NiFeOOH is deposited, the GC will degrade with time under solar OER operation (while BDD is stable) as shown in page 14 and Supplementary Figs. 14 and 15. In that case, a thicker GC sheet will obviously offer longer stability, but it is not recommended because photocurrents will be poorer anyway due to the absence of Ni/NiFeOOH catalyst.

In terms of performance, glassy carbon and boron-doped diamond are very conductive. The specific resistances ρ of GC and BDD are $45 \mu\Omega \text{ m}$ and $10 \mu\Omega \text{ m}$ (data provided by the manufacturer), respectively, so the resistances across the opposite faces of 1 cm^2 GC and BDD sheets are just 0.00045Ω and 0.00008Ω . The measured total resistance across the FTO/carbon/adhesive/GC (or BDD) interfaces were 3-4 Ω per cm^2 , as shown in Supplementary Fig. 6 and explained in page 9. Therefore, the inner resistances of GC and BDD due to their thickness are irrelevant, and they will not affect the final performance.

In view of this comment, we have added the following to the manuscript:

Page 9:

After successfully demonstrating the OER catalytic activity of the GC/Ni/NiFeOOH sheet, we focus on the PEC performance and stability of the CsPbBr₃ photoanode protected with it (Fig. 4). The protective sheet was adhered and electrically contacted to the CsPbBr₃ photo-absorber device by using a thin adhesive layer prepared by diluting a commercial adhesive in toluene and spin coating. To confirm the electrical contact across the interfaces, 2-electrode J-V scans of FTO/carbon and FTO/carbon/adhesive/GC (and later BDD) were measured, showing similar high slope values, that is, similar small resistances of only 3-4 Ω (Supplementary Fig. 6). The surface of printed carbon is rough, ensuring the presence of surface pockets for the adhesive and spikes for the direct electrical connection to the GC (and later BDD) sheet, as sketched in Supplementary Fig. 6. Therefore, no conductive fillers are needed in the adhesive layer provided there is enough roughness at the interface.²⁹ Moreover, the GC (and BDD) sheets are very conductive, 45 (and 10) $\mu\Omega \text{ m}$, so the resistance across their 1 (and 0.8) mm thickness is negligible, only 450 (and 80) $\mu\Omega$ per cm^2 sheet area. ...

Page 18:

Protective catalytic sheet fabrication

Protective sheets consist of a substrate (GC sheet or BDD sheet), catalytic layers, and adhesive layer. Before fabricating each protective sheet, the front sides of the GC sheets (JM Material CO., Ltd., 10 mm × 10 mm × 1 mm, resistivity: 45 $\mu\Omega \text{ m}$)... For the devices with BDD sheets (CVD BDD sheet, Ningbo Gh Diamond Tool CO., Ltd., 10 mm × 10 mm × 0.8 mm, resistivity: 10 $\mu\Omega \text{ m}$, one side polished),...

10. Please provide details of atomic ratio for the EDS result in figure 4 supplementary.

Response:

We thank the reviewer for the helpful suggestion. We have added the atomic ratio results to the caption of Supplementary Fig. 4 as follows:

Supplementary Fig. 4 Top-view SEM micrographs and EDS spectra of GC/Ni nanopyramids (a and b) and GC/Ni-nanopyramids/NiFeOOH (c and d). The EDS spectra show increased energy peak of Ni element in GC/Ni and increased Ni and Fe element peaks GC/Ni-nanopyramid/NiFeOOH. The atomic ratio of C, O, Ni and Fe in GC/Ni is 7.72:6.37:85.91:0 and in GC/Ni/NiFeOOH is 6.55:24.85:65.30:3.30.

11. Supplementary fig. 6 showing the resistance measurement, I wonder why the J-V plot does not meet the origin of coordinates? This is an ohmic contact and I believe the J-V characteristic should be linear and meet (0; 0).

Response:

We thank the reviewer for this question. Yes, it is an ohmic contact with a low resistance as explained in the manuscript. The deviation from the origin (0,0) is small, just 10-20 mV, much smaller than the mismatch between work functions of FTO and carbon (170 meV). Therefore, we attribute the small 10-20 mV deviation from the origin to charging/discharging of surface defects on the FTO and its interface with the carbon layer in our measurement. FTO is known to be rougher and have more surface defects than the alternative indium tin oxide (ITO) (*Nanoscale Adv.*, 2023, 5, 1492-1526), but it is much more cost-effective since indium is scarce.

In view of this comment, we have zoomed out the figure to include negative voltages and added a sentence to the caption as follows:

Supplementary Fig. 6 ... (b) J-V scans of these structures to calculate resistance presented in the inset. The area was controlled to be 1 cm² for all the samples. The small 10-20 mV deviation from the origin is assigned to charging/discharging of surface defects on the FTO and its interface with the carbon layer in this measurement.

12. The active surface area of the electrode is 0.197 cm². How about the active area (illuminated area) of the perovskite solar cells?

Response:

The illuminated area of the perovskite solar cell was the same as the reaction surface area (0.197 cm²). We used circular masks on both sides to control the illuminated and wet areas and for calculating the photocurrent density.

In view of this comment, we have added more context in (Photo)electrochemical measurements section as follows:

Page 20:

...(circular mask with predefined diameter of 5 mm was used to control the active surface area 0.197 cm²) against a Pt counter electrode and a KCl-saturated Ag/AgCl reference electrode. The illuminated area was also controlled to be the same as the active surface area by using a front circular mask.

13. Line 279, page 12 in the manuscript, the authors assigned the improved performance of BDD device compared to the GC is due to the higher surface roughness thus more reaction sites. Does that mean due to the increased surface area? Should the author provide a specific surface area measurement to support this point? Because increased roughness does not necessarily lead to increased surface area.

Response:

Yes, we meant due to increased surface area. It is well-known in electrocatalysis that a higher surface area of electrocatalyst favors catalysis, in this case for OER. We have calculated the surface area of the catalytic sheet front surface by AFM. Surface areas (S_a) of front side of GC and BDD were 1.017 m² and 1.084 m² per m² of projected area (m²m⁻²), respectively. Surface areas of GC/Ni/NiFeOOH and BDD /Ni/NiFeOOH were 1.096 m²m⁻² and 1.561 m²m⁻², respectively. The surface area of BDD/Ni/NiFeOOH is therefore larger than that of GC/Ni/NiFeOOH sheet, which provides more reaction sites for OER, so higher PEC photocurrents. This is also evident in the AFM micrographs (Supplementary Fig. 12e-f.), there are more nanopyramids structures on the BDD surface for the deposition of NiFeOOH electrocatalyst.

In view of this comment, we have added more discussion in the manuscript and Supplementary Fig. 12 as follows:

Page 12:

...The surface area S_a values of the front side of GC before and after Ni and NiFeOOH deposition are 1.017 and 1.096 m² per m² of projected area (m²m⁻²), respectively, and those of BDD are 1.084 and 1.561 m²m⁻², respectively. Therefore, the improved performance of BDD devices compared to those of GC is in part assigned to its ~~double surface roughness~~ increased surface area, which promotes charge transfer and oxygen evolution by providing more reaction sites and interface to the electrolyte.

Supplementary Fig. 12 AFM micrographs of (a) back side of GC before adhesive coating, average roughness $R_a = 0.28$ nm, (b) back side of BDD before adhesive coating, $R_a = 0.31$ nm, (c) front side of GC before Ni electrodeposition, $R_a = 45.7$ nm, surface area $S_a = 1.017$ m²m⁻², (d) front side of BDD before Ni electrodeposition, $R_a = 125.1$ nm, $S_a = 1.084$ m²m⁻², (e) GC/Ni/NiFeOOH, $R_a = 85.5$ nm, $S_a = 1.096$ m²m⁻², and (f) BDD/Ni/NiFeOOH, $R_a = 160.4$ nm, $S_a = 1.561$ m²m⁻².

14. The remarkable stability of the cells reported in this work is clearly much higher than the previous reports. However, in order to make a reasonable assessment, please provide a comparison of the photon-to-current efficiency with previous report as well?

Response:

We thank the reviewer for this comment, we agree that the photon-to-current efficiency is important to the commercial application of PEC water splitting. The photon-to-current efficiency depends on the bandgap and optoelectronic properties of the photoabsorber. Our focus in this work is on the stability of the catalytic sheets, so we used all-inorganic CsPbBr₃, known to be more stable than hybrid perovskites. Using hybrid perovskites with better ABPE such as FAPbBr₃ (ref. 30 in our manuscript) but unstable nature under irradiation would have created ambiguity and complication in the presentation of GC and BDD protective sheets.

In view of this comment and comment 2, we measured the applied bias photon-to-current efficiency (ABPE) and added it in the revised manuscript and Supplementary Information. We have also added more discussion in the manuscript and Supplementary Information as follows:

A comparison of reported stabilities for halide perovskite-based devices including photoanodes is displayed in Fig. 7. Different types of protected perovskite-based photoanodes have been reported, but most of the devices do not have stable photocurrents, due to degradation of the photoabsorber and/or the protection. They often show a significant decrease in performance during the stability test, although they can achieve a high initial ABPE due to their organic-inorganic hybrid composition and small bandgap. Details of the photoanodes are listed in Supplementary Table 1. Our all-inorganic CsPbBr₃ photoanodes protected with GC/Ni/NiFeOOH and especially BDD/Ni/NiFeOOH achieve record stabilities, that is record preservation of the initial photocurrent. Moreover, our approach follows an economical and scalable fabrication strategy using commercial sheets, spin coating and electrochemical deposition, as well as abundant elements (carbon, nickel, iron and oxygen), to achieve high stability and performance.

Supplementary Table 1 Comparison of reported photoelectrochemical devices including halide perovskite photoanodes for solar-driven OER (AM1.5G, 1 sun)

Perovskite material	Protection layer ¹	Stability duration test	J _n [#]	Initial max. ABPE	Manuscript Reference
FAPbBr ₃	GS/NiFe alloy/NiFe LDH	60 h	43%	8.5%	30
(FAPbI ₃) _{0.97} (MAPbI ₃) _{0.03}	GS/IrOx cat.	16.3 h	60% ^{\$}	11.3%	29
FA _{0.83} Cs _{0.17} Pb(I _{0.8} Br _{0.2}) ₃	Field metal/Ni	40 h	38 %	5.8%	24
FAMAPbI ₃	Ni/NiFeOOH	48 h	50 %	9.2%	26
FAMAPbI ₃	Ni foil/3D Ni ₂ NiFe	20 h	60 % [@]	8.9%	27
MAPbI ₃	Carbon nanotube /polymer	0.5 h	65 %	-	25
MAPbI ₃	Carbon	12 h	70 %	-	23
CsPbBr ₃	GS/Ir mol. cat.	27 h	80 %	-	28
CsPbBr ₃	dense GS/GS/ /NiFeOOH	70 h ^{&}	83 %	-	31
CsPbBr ₃	GC/Ni/NiFeOOH	168 h	95 %	2.5%	This work
CsPbBr ₃	BDD/Ni/NiFeOOH	210 h	97 %	3.8%	This work

15. One problem with the carbon based-PSC is the scalability due to the sheet resistance, could the author demonstrate the performance of the cell with a large active area?

Response:

We agree with the Reviewer that sheet resistance of carbon layers can limit the scale-up of carbon-based PSCs, although it can be improved by enhancing the interconnectivity of the carbon particles (Energy Environ. Sci., 2019, 12, 3437-3472.). It is also known that the poor sheet resistance of a transparent conductive oxide (TCO) can be improved with a conductive metal grid in contact with the TCO (ACS Energy Lett., 2017, 2, 9, 1978–1984, and Solar RRL, 2022, 6, 2100865.). This makes us think that adding very conductive GC or BDD sheets on top of the carbon layer may alleviate its sheet resistance, as metal grids do on TCO, and improve the scalability of the device.

Anyway, our manuscript is not focused on scale-up or carbon layer sheet resistance, but on the presentation of GC and BDD protective sheets with Ni/NiFeOOH for photoanodes. In a previous article in *Adv. Mater.* 2023 using graphite protective sheets (that are also very conductive), we already scaled the carbon-based PSC devices to 113 mm². Going larger is not easier, and it is pure engineering and out of the scope of this manuscript.

REVIEWERS' COMMENTS

Reviewer #1 (Remarks to the Author):

The authors have addressed my comments and greatly improved the paper. I do not have additional scientific corrections to request. The long term stabilities are good, but I do still think that this work shows incremental improvement rather than a ground breaking new science.

Reviewer #3 (Remarks to the Author):

My concerns have been addressed by the author. The quality and comprehensiveness of the manuscript have been improved. I have no further concerns. I recommend this manuscript to be considered for publication.

Reviewer #4 (Remarks to the Author):

I agree that the concerns raised by the Reviewer 2 have been addressed. However, from my perspective, the novelty of this manuscript is limited and it lacks sufficient scientific significance. I cannot recommend this manuscript to be published in nature communication.

Response to Reviewer Comments

“Ultrastable halide perovskite CsPbBr₃ photoanodes achieved with electrocatalytic glassy-carbon and boron-doped diamond sheets” by Zhu et al.

Manuscript submitted to *Nature Communications* (Research Article, No. NCOMMS-23-50474A)

Reviewer #4:

I agree that the concerns raised by the Reviewer 2 have been addressed. However, from my perspective, the novelty of this manuscript is limited and it lacks sufficient scientific significance. I cannot recommend this manuscript to be published in nature communication.

Response:

We are sincerely thankful to Reviewer #4 for acknowledging our efforts in addressing the concerns raised by Reviewer #2. Nonetheless, we respectfully contend that the scientific novelty and significance of our manuscript are substantial and merit publication in *Nature Communications*.

Our manuscript introduces, for the first time, an innovative approach by employing glassy carbon and boron-doped diamond as protective layers for the halide perovskite, CsPbBr₃, diverging from the conventional use of graphite or Ni sheets. This strategic selection not only marks a pioneering integration within photoelectrochemical systems but also significantly enhances the stability of water-unstable semiconductors. To our knowledge, this is the inaugural instance where these catalytically active sheets have been coupled with any semiconductor for application in photoelectrodes, underscoring the groundbreaking nature of our research.

The utilization of glassy carbon and boron-doped diamond, materials that are both readily accessible, positions our findings as a pivotal advancement for the broader adoption of halide perovskites in photoelectrochemistry. This development holds potential transformative implications for diverse applications, including but not limited to, aqueous electrolyte-based sensors.

Furthermore, our work elucidates the integration of novel Ni nanopillars and NiFeOOH onto these innovative substrates, which not only further elevates stability but also significantly enhances catalytic activity. This element of our research has not been previously explored in our prior publications nor, to the best of our understanding, in the existing literature.

Additionally, we detail a novel method for the deposition of an adhesive layer on the reverse side of the glassy carbon and boron-doped diamond sheets. This technique facilitates the secure attachment of these sheets to a photo-absorber structure, contingent upon sufficient surface roughness, further emphasizing the practicality and applicability of our proposed methodology.

In light of these considerations, we firmly believe that our manuscript presents groundbreaking findings that contribute meaningfully to the field of photoelectrochemistry and align with the esteemed standards of *Nature Communications*.